# Exploration of forestry carbon sequestration practice path in Guizhou province-based on evolutionary game model

**Wu Yang**[1,2], **Zhang Min**[1,2]*, **Cui Tao**[1,2], **Yan Jun**[1,2]

1 Guizhou Institute of Technology, School of Resources and Environmental Engineering, Guiyang, China,
2 Engineering Research Center of Carbon Neutrality in Karst Areas, Ministry of Education, Guiyang, China

* zhangmin@git.edu.cn

## Abstract

Guizhou Province has abundant forest resources, and it has great economic value and social benefits to explore the practical path of forestry carbon sequestration. Based on the current situation of forestry carbon sequestration development in Guizhou Province, this paper innovatively integrates forestry carbon sequestration indicators into the existing Environmental, Social and Governance(ESG) evaluation system using an evolutionary game model. It analyzes the factors restricting forestry carbon sequestration and explores the influencing factors of forestry carbon sequestration benefit sharing bodies in Guizhou. Through regression analysis, the paper discusses the impact of enterprise ESG scores, government subsidy amounts, and forestry carbon sequestration costs on forestry carbon sequestration purchase volume. The research results show that enterprise ESG scores and government subsidy amounts have a significant positive impact on enterprise forestry carbon sequestration purchase volume, while forestry carbon sequestration costs have a significant negative impact. The results have passed the robustness test in different industries. The simulation analysis results show that the stable point of the evolutionary game is (1,0,1) and (1,1,0), which verifies that the ESG rating system with forestry carbon sequestration integration can promote enterprises to purchase more forestry carbon sequestration, i.e., the effectiveness of forestry carbon sequestration in activating the ESG rating system mechanism. Based on the research conclusions, the paper puts forward policy implications: the government should accelerate the construction of localized ESG rating systems, improve enterprise information disclosure and supervision, increase subsidies and reduce forestry carbon sequestration costs, and optimize carbon quota design.

## 1. Introduction

Since the reform and opening up, the Chinese economy has experienced rapid growth. Energy consumption has provided support for economic growth, but it has also resulted in significant carbon emissions, leading to environmental damage caused by excessive greenhouse gas emissions. China's economic growth and carbon emissions are in a "weak decoupling" state, which

---

"Guizhou Forestry Yearbook" section and is provided as supporting information file.

**Funding:** This research was supported by Regional Project of National Natural Science Foundation of China (41463003), Surface Project (41573043), Concealed Ore Deposit Exploration and Innovation Team of Guizhou Colleges and Universities (Guizhou Education and Cooperation Talent Team [2015]56), Provincial Key Discipline of Geological Resources and Geological Engineering of Guizhou Province (ZDXK[2018]001), Huang Danian Resources of National colleges and universities Teachers' Team of Exploration Engineering (Teacher Letter [2018] No. 1), Geological Resources and Geological Engineering Talent Base of Guizhou Province (RCJD2018-3), Key Laboratory of Karst Engineering Geology and Hidden Mineral Resources of Guizhou Province (Qianjiaohe KY [2018] No. 486Guizhou Institute of Technology Rural Revitalization Soft Science Project(2022xczx10), Education and Teaching Reform Research Project of Guizhou Institute of Technology (JGZD202107,2022TDFJG01). The funders had no role in study design, data collection and analysis, decision to publish, or preparation of the manuscript.

**Competing interests:** The authors have declared that no competing interests exist.

means that there may be outdated emission reduction technologies and ineffective emission reduction management methods in China's energy consumption. With the signing of the Paris Agreement and China's increasing carbon emissions, China is facing heavier international pressure to reduce greenhouse gas emissions. As a "responsible major power," China has been making its own efforts and contributions to addressing climate change, actively exploring and trying to establish a carbon emissions trading market to promote the low-carbon emission of high-emission enterprises and suppress the continued rise of domestic carbon emissions using market means [1–5] (Fig 1).

In order to scientifically reduce the emissions of high-carbon enterprises, China has designed from the macro and micro levels. At the macro level, the regulatory authorities will include high-carbon enterprises in the scope of emission control, through accelerating the construction of carbon market, building a scientific and orderly carbon trading system, with the help of market mechanism to achieve carbon emission reduction [6–10]. In the process of improving the construction of carbon market, China Certified Emission Reduction (CCER) project trading market is an important way to reduce emissions in the carbon market. It was officially restarted under the document "Measures for the Management of Voluntary Greenhouse Gas Emission Reduction Trading (Trial Implementation)" issued by the Ministry of Ecology and Environment in 2023. As an indispensable carbon offset product in the CCER market, forestry carbon sequestration mainly uses forests to absorb and fix carbon dioxide through afforestation, forest management and other activities, which has significant advantages in ecological benefits, is an important innovative way to achieve the goal of carbon neutrality, and faces new development opportunities. However, in the actual CCER market, the purchase demand of enterprises for forestry carbon sinks is insufficient, and the development of forestry carbon sinks is restricted. At the micro level, the regulatory authorities urge high-carbon enterprises to transform and increase efficiency through mandatory disclosure of environmental information. Among them, Environmental, Social and Governance)(ESG)is a new investment concept and evaluation tool in recent years, covering the three dimensions of environmental, social and corporate governance information. With the help of ESG information disclosure and rating system, investors can evaluate the comprehensive operation and sustainable development ability of an enterprise in a multi-dimensional and all-round way, and then influence the decision-making of the enterprise. Qiu [11] proposed that good ESG performance can ease the financing constraints of enterprises; Li [12] believed that a complete ESG rating system is an important starting point to achieve the "double carbon" goal; Hu [13] emphasized that ESG rating can significantly promote the green transformation of enterprises through market incentives and external supervision mechanisms. To sum up, ESG has the function of reducing financing costs and enhancing enterprise value, which is of great significance to promote emission reduction of high-carbon enterprises.

The above macro-level and micro-level designs are effective ways to promote emission reduction of high-carbon enterprises, but they often play an independent role and do not establish an effective linkage mechanism. As far as the carbon market mechanism is concerned, due to the sufficient supply of market quotas and the relative price advantage of excess carbon quotas, emission control enterprises usually tend to choose to purchase excess quotas to offset excess emissions, rather than forestry carbon sequestration projects to achieve carbon sequestration at the ecological level. Therefore, how to promote high-carbon enterprises to purchase forestry carbon sinks spontaneously from the market mechanism is of great significance to the realization of the goal of carbon neutrality. In order to solve the problem of insufficient demand for forestry carbon sinks, ESG may become an innovative way to promote emission reduction of high-carbon enterprises at the micro level. Qian [14] pointed out that ESG has the ability to guide the flow of funds to green low-carbon areas. It can be seen that an

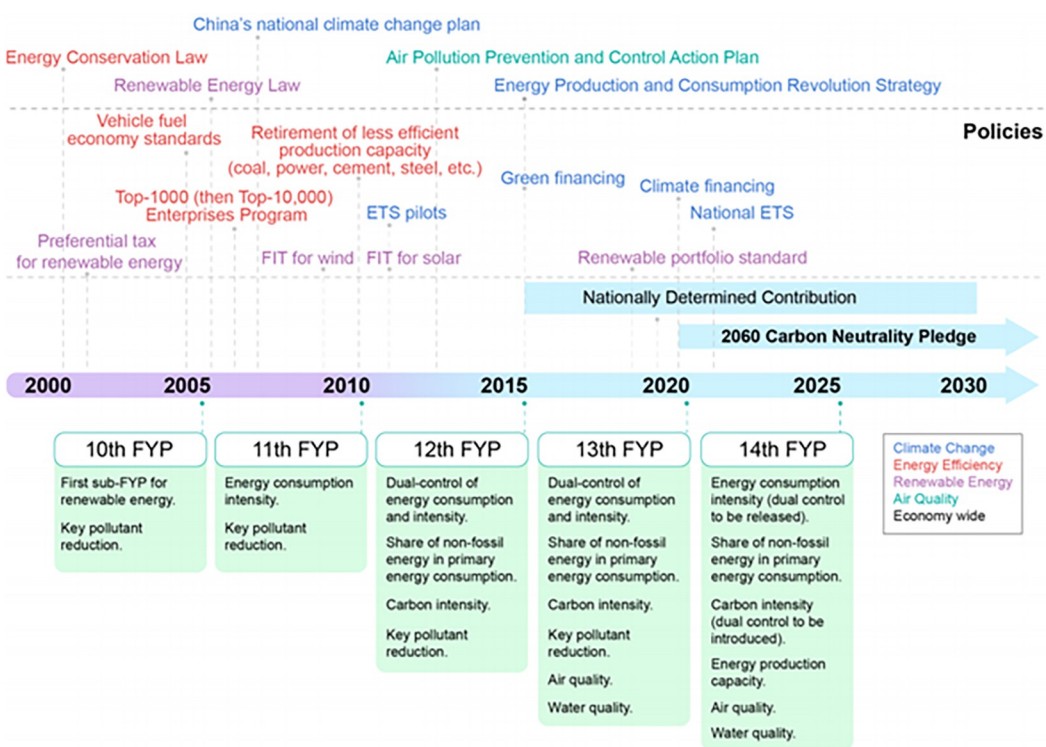

**Fig 1. China's energy and climate policy plan.**

effective ESG mechanism can guide high-carbon enterprises to buy more forestry carbon sinks.

According to China's dual carbon policy, carbon emitting enterprises face rigorous post-graduate entrance examinations. By 2060, as coal-fired power plants and coal based industrial processes that have not adopted emission reduction measures have been basically eliminated, the proportion of coal combustion related emissions will be reduced by about 50% compared to 2020. During the period of 2021–2060, process emissions (inherent emissions generated by chemical reactions in industrial processes) will decrease by about 90%, and the proportion of total emissions will almost double, due to the fact that it is extremely difficult to eliminate process emissions in certain heavy industry sectors, especially the cement and steel industries. The remaining emissions of the energy system by 2060 will be fully offset by negative emissions generated by BECCS and direct air capture and storage. In China's efforts to achieve full economic greenhouse gas neutrality before 2060, carbon removal technology can also be used to offset some of the more difficult to reduce non carbon dioxide greenhouse gases. Therefore, the carbon sequestration capacity of ecosystems, especially forestry carbon sequestration, is particularly important (Fig 2).

Guizhou Province is located in western China and has state-owned forest areas, ranking high among all provinces in terms of forest area. After the implementation of the comprehensive ban on logging natural forests in state-owned forest areas, the accumulation area and quality of forests in Guizhou Province have been significantly improved, providing a unique natural resource advantage for the development of forestry carbon sinks in Guizhou Province. However, according to the data from China's voluntary emission reduction trading information platform, the development potential of forestry carbon sinks in Guizhou Province has not been fully activated, and the supply of forestry carbon sinks is relatively insufficient [14–18].

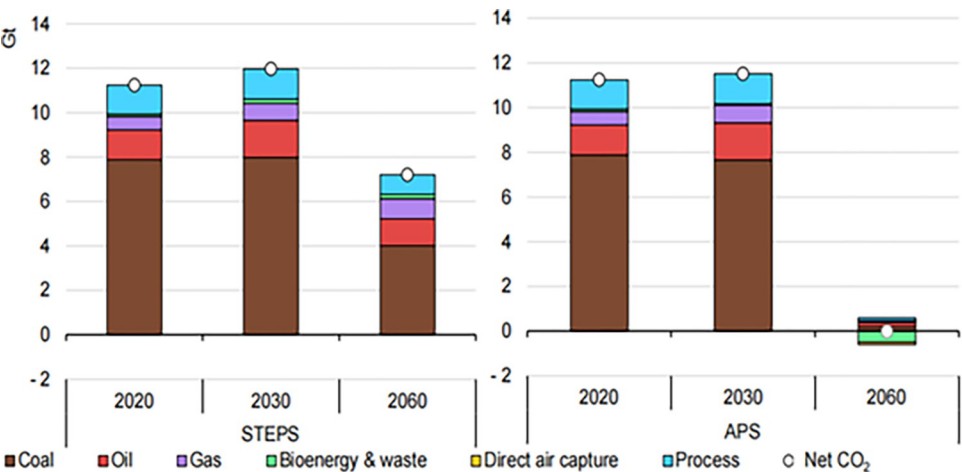

**Fig 2. $CO_2$ emissions in the industrial sector.**

As of 2023, the implementers of forestry carbon sequestration projects in Guizhou Province are all local forestry bureaus. This indicates that the supply subject of forestry carbon sink in Guizhou Province is relatively single. Forestry carbon sequestration projects have the characteristics of large initial investment, long cycle, and difficulty as collateral for mortgage loans. The single supply subject of forestry carbon sink will increasingly constrain the development of forestry carbon sink in Guizhou Province, affecting the stability of effective supply of forestry carbon sink in Guizhou Province, and thus affecting the realization of ecological and social benefits of forestry carbon sink. Compared to Guangdong Province's carbon inclusive mechanism and other forestry carbon sink development policies, Guizhou Province currently does not have a systematic forestry carbon sink development policy and support system. This hinders the improvement of the external environment for the development of forestry carbon sinks in Guizhou Province, affects the stability of economic benefits that forestry carbon sink suppliers can obtain, and is not conducive to improving the enthusiasm of forestry operators to provide forestry carbon sinks, thereby affecting the effectiveness and stability of forestry carbon sink supply in Guizhou Province, leading to a vicious cycle in which the ecological and social benefits of forestry carbon sinks are difficult to achieve. When forestry management enterprises and governments form a forestry carbon sink benefit sharing body, that is, when a "cooperative win-win" model of forestry carbon sink is formed, it can effectively improve the stability of effective supply of forestry carbon sink, achieve the ideal cycle of ecological, social and economic benefits of forestry carbon sink, and promote the sustainable development of forestry carbon sink in Guizhou Province [19–24].

In summary, this paper takes the current development status of forestry carbon sinks in Guizhou Province as the starting point, analyzes the behavioral characteristics of stakeholders in forestry carbon sinks in Guizhou Province, constructs an evolutionary game model to analyze the influencing factors of the construction of forestry carbon sink benefit sharing bodies in Guizhou Province, and proposes countermeasures and suggestions to promote the stable and innovative development of forestry carbon sinks in Guizhou Province based on the analysis results. On the one hand, this can fully and effectively utilize the forest resources in Guizhou Province to promote the industrial and orderly development of forestry carbon sinks, accelerate the speed of China's greenhouse gas emissions reduction, and contribute to the development of the national ecological economy. On the other hand, this can broaden the ways of ecological civilization construction in Guizhou Province, provide new economic development

channels for state-owned forest areas, and promote the sustainable development of forestry carbon sinks in Guizhou Province.

## 2. Literature review

First of all, combing the development status of ESG rating system adopted by domestic and foreign emission control enterprises, and pointing out the existing problems, laying the foundation for this innovative construction of operational mechanism; Furthermore, by sorting out the domestic and foreign literature on the realization of ESG functions, we focus on the behavior and connection of multiple subjects in the operation of ESG, which provides a theoretical reference for building and testing the evolutionary game model of the effectiveness of the mechanism.

### 2.1. Status quo of ESG rating system

At present, China has initially formed an ESG information disclosure system and mechanism, which is promulgated by the government and assisted by financial institutions. However, due to the lack of a unified standard for the ESG information disclosure system of listed companies, especially with the proposal of the "double carbon" target, the content of disclosure is required to increase gradually, and the form of disclosure is more standardized. Enterprises are facing more severe ESG management challenges and need to make more efforts. Relevant foreign studies such as Escrig-Olmedo et al. [25] pointed out that ESG institutions and sustainable development index currently use a variety of methods and lack of standardization. Dorfleitner et al. [26] also considered that the concept of ESG measurement was obviously inconsistent; Eccles et al. [27] discussed the differences among ESG rating agencies and pointed out that these differences would lead to inconsistent rating results; Gibson et al. [28] mentioned that the lack of uniform standards leads to poor comparability of ESG rating results among different institutions. In addition, there are significant differences in the models and data sources used by rating agencies such as Sustainalytics and MSCI in the evaluation process. These problems make it difficult for investors to accurately assess the ESG performance of enterprises.

Abhayawansa et al. [29] believe that some emission control enterprises may be unwilling or unwilling to fully disclose the real ESG performance, and the data sources, weights and methods lack transparency. A considerable number of emission control enterprises selectively disclose information that is beneficial to their own image, while concealing the situation that is not beneficial to themselves, which affects the reliability and transparency of ESG rating. Most of the emission control enterprises are heavy pollution enterprises, and the quality of information disclosure in the environmental dimension is particularly important, but there are great differences in the disclosure under the existing rating system. However, due to the characteristics of heavy pollution industries, there are still some common characteristics in the indicators disclosed by various emission control enterprises. Based on this, this paper takes the localization innovation of ESG rating system of emission control enterprises under the goal of "double carbon" as the research direction, focusing on the environmental dimension of ESG rating system, and selecting unified environmental indicators for emission control enterprises.

### 2.2. Multi-subject behavior and connection of ESG function realization

There are many researches on the issue of multiple agents in ESG function implementation mechanism, and different opinions have been formed.

(1) Enterprises: In recent years, more and more enterprises have begun to incorporate ESG principles into their strategic planning and operations to address growing environmental and social challenges. Zhang [30] systematically combed the supporting theory of ESG information

disclosure research of listed companies, and further revealed that there is a significant positive correlation between corporate environmental, social and corporate governance information disclosure and corporate development. Li [31] verified that ESG performance and its three dimensions can significantly improve enterprise performance and innovation level. Wang [32] analyzed the different channels of ESG performance to realize the value effect, one is to reduce the financing cost and promote the book value of the enterprise; the other is to enhance the market attention, thereby enhancing the market value of the enterprise.

(2) Government: Wang [33] believe that the government plays a vital role in promoting ESG practice. In order to achieve the vision of "double carbon" goal, the government has launched various measures to promote enterprises to actively implement ESG. Chen [34] pointed out that the government can distinguish enterprises according to their internal ESG conditions, the level of ESG development and the content of ESG actions being implemented, and design targeted boosting methods. Huang [35] pointed out that the central environmental protection supervision played a more obvious role in promoting the ESG performance of non-state-owned enterprises and heavily polluting enterprises, which could significantly enhance the level of active risk-taking, increase the intensity of environmental subsidies and enhance the momentum of green technological innovation of enterprises, thus improving the ESG performance level of enterprises. Meng [36] studied the impact of government tax incentives on corporate ESG and found that when capital market regulation is high or corporate financial redundancy is abundant, tax incentives have a stronger effect on corporate ESG performance.

(3) Investment institutions: Freya Williams [37] pointed out that corporate social responsibility activities are driven by investors with a long-term investment vision because it takes time to build a reputation; Kim [38] also argued that investors tend to pay attention to ESG concepts out of social preferences and altruistic motives; Tang [39] pointed out that investors themselves have the characteristics of self-discipline and community, which determines that after self-reflection, investors find that ESG is their own constitutive purpose and the common purpose of investor groups, thus pursuing the concept of ESG. Based on the above motivation, investment institutions are important stakeholders of ESG. Chen Xiao [40] pointed out that ESG investment concept can promote enterprises to increase investment in the field of climate change, help financial institutions to avoid climate change risks, and promote regulatory policies to guide the flow of funds to the field of climate governance.

In conclusion, the existing body of research concerning the role of multiple agents in the operational mechanisms of Environmental, Social, and Governance (ESG) function realization is notably extensive, offering substantial literature support for this study. However, certain limitations persist: from a research perspective, current literature predominantly emphasizes the evaluation of the direct outcomes and advantages of ESG practices, while there is a lack of comprehensive analysis regarding the interaction mechanisms among various stakeholders. Additionally, in terms of research methodology, there is insufficient exploration into strategies for enhancing ESG practices, particularly in motivating and guiding emission control enterprises to elevate their ESG performance. Most studies tend to concentrate on macro-level policy analysis, neglecting a thorough examination of micro-level corporate decision-making behaviors and their underlying mechanisms. Furthermore, there is a scarcity of research that systematically integrates ESG practices with carbon market mechanisms to evaluate their collective impact on achieving carbon neutrality. Consequently, it remains imperative to investigate how to further activate the ESG function to effectively contribute to the realization of the "double carbon" goal. Additionally, exploring ways to stimulate market demand for forestry carbon sequestration and promote afforestation through market mechanisms and policy initiatives is essential for attaining carbon neutrality. These issues warrant in-depth study and discussion. This paper aims to construct a coupling mechanism between forestry carbon

sequestration and the ESG system, utilizing an evolutionary game model for simulation to assess the mechanism's effectiveness, thereby providing a theoretical foundation and practical guidance for enhancing the ESG rating system of emission control enterprises in Guizhou and facilitating the green transformation and sustainable development of high energy-consuming industries.

## 2.3. The development of evolutionary game theory and model application cases

The core of evolutionary game models is to replicate dynamic equations and evolutionarily stable strategies (ESS). Copying dynamics refers to the growth rate of the proportion of people who select a certain strategy being equal to the difference between the payment obtained by that strategy and the average payment [41,42]. By constructing replicated dynamic equations, the evolutionary trajectory of an evolutionary game system to reach a stable equilibrium state can be derived. Evolutionary Stability Strategy (ESS) is an equilibrium strategy point that maintains a stable state. In game systems, evolutionary stable strategies are always in a stable state. This means that even if some individuals in the group experience occasional deviations, the strategy will still be in equilibrium after replicating dynamics. This paper uses an evolutionary game model to analyze the evolutionary stability strategies and paths of stakeholder games in forestry carbon sequestration, and then dissects the specific influencing factors of the construction of forestry carbon sequestration benefit sharing entities. The Stanford University Research Institute first defined the concept of "stakeholders" in 1963, stating that stakeholders are the group of people outside of the enterprise who can profit from and influence the operation of the enterprise. The research of American economist Freeman has expanded the connotation of stakeholders. In 1984, Freeman's book "Strategic Management: A Stakeholder Approach" constructed a systematic framework for stakeholder theory, marking the transformation of "stakeholders" from a concept to a comprehensive theoretical system. Freeman believes that stakeholders refer to any individual or group who can influence the achievement of organizational goals and whose behavior is also influenced by organizational goals. At this point, individuals or groups such as communities and governments, in addition to enterprises, have also been included in the research scope of stakeholder theory [43–45]. Government agencies are the makers of policies related to forestry carbon sinks, and their behavioral decisions to a certain extent affect the demand for forestry carbon sinks in society and the realization of economic benefits of forestry carbon sinks, thereby affecting the stability of effective supply of forestry carbon sinks. As the direct implementers of forestry carbon sequestration projects, the management decisions of forestry operators directly affect the effective supply of forestry carbon sequestration, thereby affecting the realization of ecological and social benefits of forestry carbon sequestration. The degree of realization of ecological, social, and economic benefits of forestry carbon sinks will also affect the willingness of forestry operators to provide forestry carbon sinks and the enthusiasm of the government to support the development of forestry carbon sinks. The emergence of independent third-party carbon sink measurement institutions is conducive to ensuring the "commodification" of carbon sinks, promoting the development of market-oriented trading of forestry carbon sinks, and ultimately facilitating the realization of ecological, social, and economic benefits of forestry carbon sinks. When the effective supply of forestry carbon sinks is insufficient or its effective supply lacks stability, third-party carbon sink measurement institutions lack development driving force and are difficult to achieve long-term development. The emergence and development of third-party carbon sink measurement institutions depend to some extent on the behavioral choices of the government and forestry operators. Therefore, in this paper, forestry carbon sink stakeholders

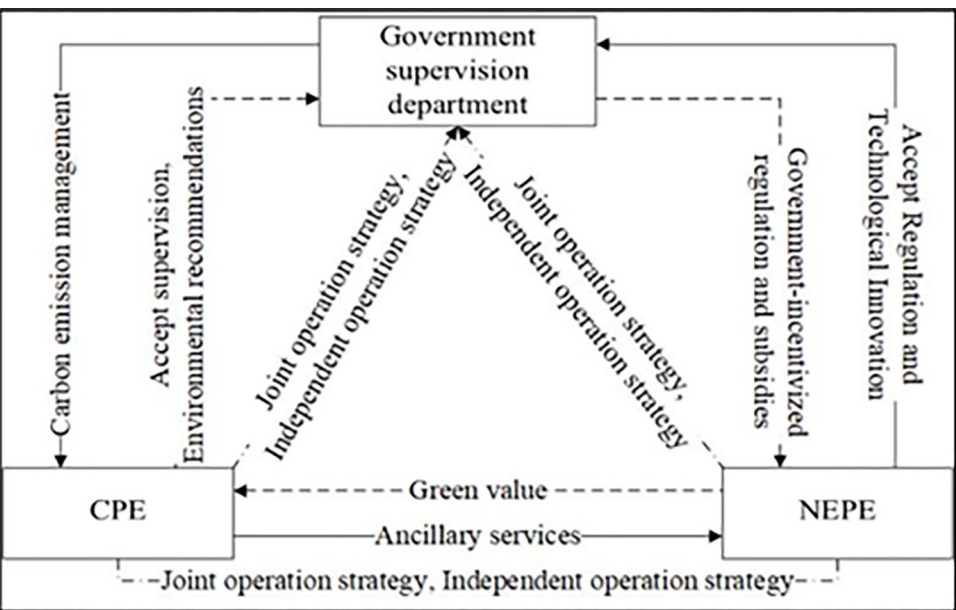

**Fig 3. Logical relationship diagram of evolutionary game model.**

refer to government agencies and forestry operators who have the most significant interest relationship with forestry carbon sinks (Fig 3).

## 3. Theoretical analysis framework and research methods

In order to further study the operation mechanism of ESG rating system for forestry carbon sequestration and emission control enterprises, this paper first reconstructs the ESG rating system for emission control enterprises, then constructs an empirical model to analyze the influencing factors of forestry carbon sequestration demand, and finally tests the effectiveness of the mechanism with the help of simulation from the perspective of non-cooperative game.

### 3.1. Theoretical analysis framework

Under the background of global climate change, achieving the goal of "double carbon" has become an important direction of China's green and low-carbon development. As an important way of carbon offset, forestry carbon sequestration absorbs and fixes carbon dioxide by means of afforestation and forest management, which has significant advantages in ecological environment protection and sustainable development. Therefore, this paper incorporates forestry carbon sequestration indicators into the ESG rating system of emission control enterprises, and tests the effectiveness of this innovative operation mechanism and analyzes its logical operation mechanism by building an evolutionary game model among enterprises, governments and investment institutions.

**3.1.1. Operation mechanism analysis.** ESG rating system plays an important role in easing financing constraints and reducing financing costs. Good ESG performance not only reflects the positive response of enterprises to energy saving and emission reduction, but also means that enterprises have strong sustainable development ability. In order to achieve their own development, enterprises tend to improve their ESG performance, thereby reducing financing costs and obtaining more investment opportunities. Therefore, the indicators in the ESG rating are important for the green transformation of enterprises.

Laws, regulations, and developmental strategies serve a crucial role in guiding actions. The integration of forestry carbon sinks into the ESG rating framework allows for the utilization of ESG ratings as a mechanism to connect businesses, governmental bodies, and investment firms, thereby establishing a novel operational framework. In particular, as companies seek to enhance their ESG ratings, they are likely to increase their acquisition of forestry carbon sinks to achieve superior ESG scores and attract investment. This approach not only fosters the sustainable development of businesses but also contributes to carbon sequestration at an ecological level, thereby aiding in the attainment of carbon neutrality objectives.

Within this operational framework, the government incentivizes companies to invest in forestry carbon sinks through policy support and financial subsidies, while also overseeing the disclosure of ESG-related information by these enterprises to ensure its accuracy and legitimacy. Investment firms base their investment choices on the ESG ratings of companies, opting to invest in those with strong ESG performance to mitigate investment risks and secure long-term returns. The outcomes of this model will illustrate the significance and impact of forestry carbon sequestration as an ESG rating criterion in facilitating the green transformation of businesses, lowering financing costs, enhancing corporate social responsibility, and advancing the achievement of carbon neutrality goals. Furthermore, it will validate the practical applicability of this approach in the realms of green finance and sustainable development, providing both a theoretical foundation and practical insights for the advancement of Guizhou's forestry carbon sequestration market and the realization of carbon neutrality objectives.

Nationalization or expropriation is an extreme measure taken by resource countries in the game of key mineral resources. In May 2019, the Zambian government announced that India's Vedanta Resources Company had violated the license terms by appointing a liquidator to take over the Konkola copper mine assets under its control, in preparation for finding new investors, citing illegal production cuts and layoffs of workers. The Zambian government and Vedanta Resources Company have long had grudges, with significant differences on issues such as electricity bills, wages, and taxes. Vedanta Resources submitted an arbitration to a South African court, which ruled that the Zambian government should cease liquidation. In April 2020, the Zambian government threatened to revoke the license of Glencore's Mopani copper mine. This highlights the Zambian government's policy intention to strengthen control over strategic mining assets.

**3.1.2. Mechanism of evolutionary game model.** Before constructing the game model among the government, emission control enterprises and investment institutions, it is necessary to understand the roles of each party in the forestry carbon sequestration market and their relationships. Through policy guidance and supervision, the government controls the actual purchase behavior of emission control enterprises, and investment institutions through capital investment, which jointly affect the operation and development of the carbon sequestration market. This paper chooses the evolutionary game model to analyze the interaction mechanism of these three parties, mainly based on two considerations.

(1) Through policy interpretation and literature analysis, it is found that the government encourages emission control enterprises to offset carbon emissions by purchasing forestry carbon sinks. This is because compared with the way of purchasing surplus carbon quotas in the market, the way of forestry carbon sequestration has achieved the effect of carbon sequestration at the ecological level, which is more conducive to promoting the realization of China's "double carbon" goal. In order to encourage enterprises to purchase forestry carbon sequestration spontaneously from the level of market mechanism design, this paper incorporates forestry carbon sequestration indicators into the existing ESG rating system, and innovatively completes the reconstruction of ESG system of domestic emission control enterprises. Under the new ESG system, the main bodies involved in the transaction of forestry carbon

sequestration market mainly include the government, emission control enterprises and investment institutions, and there will inevitably be a collision of interests among them. In order to obtain higher ESG scores to attract investment from investment institutions, enterprises choose to purchase more forestry carbon sinks, but at the same time, enterprises also face the cost-effectiveness of purchasing forestry carbon sinks, that is, they need to weigh the benefits from investment institutions after purchasing forestry carbon sinks and the cost of maintaining existing purchasing strategies. In order to let enterprises choose to buy more forestry carbon sinks, the government supervises the ESG rating information of enterprises, which will limit the profits and profit models of emission control enterprises, affect the operating costs of enterprises, and will inevitably increase the management costs of the government. Investment institutions choose to invest in emission control enterprises with higher ESG scores, which inevitably requires the government to enhance the supervision of ESG information disclosure, otherwise investment institutions will face greater investment risks, while emission control enterprises that do not purchase forestry carbon sinks will also bear economic losses and reputation losses. Accordingly, how does the government adjust its supervision to ensure the balance between the profit of emission control enterprises, the risk benefit of investment institutions and the goal of promoting the purchase of forestry carbon sinks, and how does the emission control enterprises choose the purchase strategy to maximize profits while promoting the realization of the government's "double carbon" goal? As well as how the investment institution adopts the investment strategy to both guarantee the investment income and positively promote enterprise's sustainable development are the questions which the tripartite gambling main body must ponder.

(2) The research and analysis of interviews reveal a competitive dynamic among government entities, emission control companies, and investment institutions. However, due to the nascent stage of the ESG system, the outcomes of this competition are suboptimal. A visit to key developers of forestry carbon sequestration in China, specifically Guangzhou Guolin Carbon Investment Eco-Technology Co., Ltd., highlighted the need for improved policy transparency within the forestry carbon sequestration market. The domestic ESG rating system has not been fully established, resulting in inadequate disclosure of ESG rating information by the government for regulatory oversight. Consequently, many emission control companies opt to mitigate excess carbon emissions by acquiring surplus carbon quotas from the market instead of investing in forestry carbon sinks. Furthermore, developers of forestry carbon sinks indicated that the current market is saturated with surplus carbon quotas. The recent reactivation of the CCER has not yet introduced new declaration channels, and there is a limited number of forests that meet the existing CCER methodology, leading to a scarcity of forestry carbon sink projects. The price of forestry carbon sequestration, approximately 152 yuan per ton, is generally higher than that of enterprise carbon quotas, which are around 100 yuan per ton. Therefore, from an economic perspective, emission control companies are likely to prioritize purchasing carbon quotas driven by self-interest. This indicates that the current carbon sequestration market mechanism fails to effectively incentivize companies to invest in forestry carbon sequestration for achieving carbon reduction goals. Thus, it is essential to innovate the existing ESG system, leveraging the influence of the ESG rating system on corporate decision-making to enhance the tripartite interaction and maximize overall benefits.

## 3.2. Research hypothesis

The research hypothesis not only helps to clarify the relationship between the variables in the model, but also verifies whether the hypothesis is valid through empirical analysis. By putting forward specific assumptions, this paper further refines the impact of ESG scores, government

subsidies and other key factors on promoting enterprises to purchase forestry carbon sinks, laying the foundation for subsequent empirical analysis and simulation research, in order to reveal the mechanism and effect of various factors in the actual operation.

**3.2.1. Research hypothesis.** A company's ESG score is a comprehensive measure of its performance in terms of sustainability and social responsibility. A high ESG score usually means that the company has performed well in environmental protection, social responsibility and governance structure. According to the existing research, the sustainable development behavior and good ESG performance of enterprises can enhance their market image, enhance investor confidence, and then improve their market competitiveness. The purchase of forestry carbon sequestration by enterprises is a concrete manifestation of their green investment behavior, and a higher ESG score can encourage enterprises to invest more in environmental protection.

On the one hand, good ESG performance can attract more green investment and reduce financing costs. Studies have shown that enterprises with high ESG scores are more likely to be favored by investors because they are regarded as more sustainable and long-term investment value. On the other hand, enterprises with high ESG scores are more inclined to fulfill their social responsibilities and take the initiative to manage carbon emissions and protect the environment. This positive environmental behavior not only enhances the market image of enterprises, but also reduces environmental risks and compliance costs. Therefore, there should be a positive relationship between ESG scores and the potential forestry carbon sink acquisitions by enterprises, so hypothesis $H_1$ is put forward.

$H_1$: The ESG performance of enterprises has a significant positive impact on the amount of forestry carbon sequestration purchased by enterprises.

Government subsidy is an important policy tool for the government to encourage enterprises to carry out environmental protection and green investment through financial means. Government subsidies can reduce the cost of green investment and improve the return on investment, thus encouraging enterprises to buy more forestry carbon sinks. According to existing research, government subsidies have a significant effect on promoting green investment and environmental behavior of enterprises. For example, government financial support can effectively reduce the expenditure of enterprises on environmental projects, thus encouraging more enterprises to participate in environmental protection projects.

Government subsidies can not only directly reduce the economic burden of purchasing forestry carbon sinks, but also enhance the environmental awareness and social responsibility of enterprises, thus promoting more environment-friendly investment. In addition, government subsidies can also play a demonstration effect, encourage other enterprises to participate in green investment, and further promote the sustainable development of the whole industry. It is found that the incentive effect of government subsidies on corporate environmental investment is particularly significant in highly polluting industries. Therefore, government subsidies should have a significant role in promoting the purchase of forestry carbon sinks by enterprises, so the hypothesis $H_2$ is put forward.

$H_2$: Government subsidies significantly promote the purchase of forestry carbon sinks by enterprises.

**3.2.2. Evolutionary game parameter assumptions.** After determining the evolutionary game model to test the mechanism, this paper makes parameter assumptions on the behavior of the three parties. (1) The behavior strategy set of the emission control enterprise $S_1$ = {$K_1$ take, $K_2$ do not take}. "Take" means that the enterprise chooses to purchase more forestry

carbon sinks under the background of the new ESG system in order to obtain a higher ESG rating; "Do not take" means that the enterprise does not make any changes according to the existing carbon sink market purchase strategy. (2) The government's behavior strategy set $S_2 =$ {$M_1$ regulation, $M_2$ non-regulation}. "Regulation" means that the government invests a certain amount of manpower, material and financial resources to the ESG rating of enterprises.Supervise the information disclosure, subsidize the emission control enterprises that are more active in disclosing ESG information, and punish the emission control enterprises that are not active in disclosing ESG information; "non-regulation" means that the government does not take any measures to intervene whether the enterprises disclose ESG information. (3) The behavioral strategy set of the investment institution $S_3 =$ {$I_1$ invest, $I_2$ do not invest}. "Investment" means that the investment institution increases investment in enterprises with higher ESG ratings under the new ESG system, and reduces investment or "does not invest" in enterprises with lower ESG ratings. It is assumed that in the initial stage of the game among the three groups of emission control enterprises, the government and investment institutions, the probability of the emission control enterprises choosing the strategy of "taking" is X, and the probability of choosing the strategy of "not taking" is 1-x; the proportion of the government choosing the strategy of "regulating" is y, and the proportion of choosing the strategy of "not regulating" is 1-y; The proportion of investment institutions choosing the "investment" strategy is Z, and the proportion of investment institutions choosing the "no investment" strategy is 1-z. Where $0 \leq X \leq 1, 0 \leq y \leq 1, 0 \leq Z \leq 1$.

## 3.3. Research methods

On the basis of the research hypothesis, this paper calculates the weight of the new ESG rating index to further verify the key role of forestry carbon sequestration in the ESG function under the new system, and then proves the effectiveness of the operation mechanism again by building an evolutionary game model from the perspective of carbon market simulation.

**3.3.1. ESG rating index weight determination.** In order to further ensure the scientificity and fairness of the operation mechanism, it is very important to determine the weights that help to clarify the relative importance of each index in the ESG performance of enterprises. Considering the effectiveness of the operation mechanism, this paper pays special attention to the weight of forestry carbon sequestration purchase in the system, and verifies its key role in promoting the green transformation of enterprises and achieving the goal of carbon neutrality by accurately measuring its influence in ESG rating. Therefore, this paper uses Analytic Hierarchy Process (AHP) to complete the determination of index weight through expert scoring method.

(1) Construct a hierarchical model. In this paper, the reconstructed ESG rating system is divided into four levels: the first level is the target level, which is the overall ESG performance of the company, and the other three levels are the factors affecting the ESG rating of the company; the second level is the first-level indicators, which are the three dimensions of environment (E), society (S) and governance (G); the third level is the second-level indicators, which are the specific indicators under the three dimensions; The fourth layer is the third-level index, which is the further refinement of the second-level index.

(2) establish a judgment matrix. Due to the different importance of the indicators in the ESG evaluation system of different industries, it is necessary to consider the nature of different industries and the external environment, based on the different scores of each layer of indicators, and finally obtain different weights and assignments for enterprises in different industries. In this paper, the Delphi method is used to construct the judgment matrix, which is a method of scoring after the expert group compares and evaluates the importance of each

**Table 1. Index importance scale of judgment matrix.**

| Scale | Meaning |
|---|---|
| 1 | Indicates that two elements are of equal importance |
| 3 | Indicates that the former is slightly more important than the latter when compared to two elements |
| 5 | Indicates that one of the two elements is significantly more important than the other |
| 7 | It means that the former is more important than the latter when comparing two elements. |
| 9 | Indicates that the former is more strongly important than the latter when comparing two elements. |
| 2, 4, 6, 8 | Indicating an intermediate value of the adjacent judgment |
| Reciprocal of 1 to 9 | Indicates the significance of the comparison of the exchange order of the corresponding two elements. |

index, and the index with high importance gets high score. The importance scale of judgment matrix indicators is shown in **Table 1**. The judgment matrix of the first, second and third-level indicators is constructed respectively, and each expert compares and scores the relative importance of the indicators under the corresponding level, so as to determine the weight of the indicators.

(3) calculate an eigenvector and an eigenvalue. Find the corresponding feature vector W for each judgment matrix constructed in this paper, as shown in Formula (1).

$$AW = \lambda_{\max} \tag{1}$$

In the Formula (1), A is a judgment matrix, which is a square matrix of n × n, in which the element aij represents the evaluation value of the relative importance of the ith element and the jth element. W is a column vector containing n weight coefficients Wi, representing the importance relative to other elements. λ is the largest eigenvalue of the judgment matrix A.

The weight is calculated according to the existing judgment matrix, and the consistency test is carried out after the result is obtained by the sum-product method to judge whether the matrix is established and whether the weight is effective. The judgment matrix is normalized to obtain a weight coefficient, as shown in Formula (2).

$$w_i = \sum_{j=1}^{n} w_j, \ (i = 1, 2, \ldots, n), \ w = (w_1, w_2, \ldots, w_n)^T \tag{2}$$

In the Formula (2), wi represents the weight coefficient of the ith element, WJ represents the weight coefficient of the jth element, and w represents a vector composed of the weight coefficients of the respective elements or indexes;.For the convenience of comparison and analysis, the sum of the components of the weight vector is equal to 1; T denotes the transpose of a matrix.

Perform consistency checks. Afterwards, a consistency test will be conducted, and the consistency indicators are shown in Eqs (3) and (4).

$$C_I = (\lambda_{\max} - n)/(n-1) \tag{3}$$

$$C_R = C_I/R_I \tag{4}$$

In Eqs (3) and (4), CI is the consistency index used to evaluate the consistency of the judgment matrix; CR is a random consistency ratio introduced considering the influence of n, where RI is the average random consistency index with a fixed value. If the consistency of the judgment matrix is better, the value of CR will be smaller; Usually, when CR ≤ 0.10, it means that the above judgment matrix has passed the consistency test; If CR>0.10, it indicates that

**Table 2. Behavior strategy combination and return matrix of emission control enterprises, governments and investment institutions.**

| Combination of strategies | Income of emission control enterprises | Government revenue | Income of investment institutions |
|---|---|---|---|
| $(K_1, M_1, I_1)$ | $E_1 + S_1 - C_1$ | $E_2 - C_2 - S_1 - S_2$ | $E_3 + S_2 - C_3$ |
| $(K_1, M_1, I_2)$ | $S_1 - C_1$ | $E_2 - C_2 - S_1$ | 0 |
| $(K_1, M_2, I_1)$ | $E_1 - C_1$ | $E_2 - S_2$ | $E_3 + S_2 - C_3$ |
| $(K_1, M_2, I_2)$ | $-C_1$ | $E_2 - S_2$ | 0 |
| $(K_2, M_1, I_1)$ | $E_4 - G_1$ | $G_1 - C_2 - S_2$ | $E_5 + S_2 - C_4$ |
| $(K_2, M_1, I_2)$ | $-G_1$ | $G_1 - C_2$ | 0 |
| $(K_2, M_2, I_1)$ | $E_4$ | $-S_2$ | $E_5 + S_2 - C_4$ |
| $(K_2, M_2, I_2)$ | 0 | 0 | 0 |

the above judgment matrix has not passed the consistency test and does not have consistency. At this time, it is necessary to adjust the judgment matrix appropriately and conduct analysis and verification again. The RI standard values are 0, 0.49, 0.84, 1.15, 1.25, 1.34, 1.41, 1.45, 1.49. Since all constructed judgment matrices have passed the consistency test, the relative weight vector W of each evaluation index in the judgment matrix can be used as the corresponding weight of each evaluation index. And by calculating the weight of the lower level evaluation indicators on the overall evaluation objective based on the weight of the upper level evaluation indicators, establish an ESG rating system based on the AHP method

**3.3.2. Construction of evolutionary game model.** In order to further test the effectiveness of the mechanism, this paper constructs an evolutionary game model to simulate the decision-making of the three parties. According to the behavior strategies of emission control enterprises, government and investment institutions, it can be concluded that there are eight game combinations among them, namely ($K_1$ adopts, $M_1$ regulates, $I_1$ invests), ($K_1$ adopts, $M_1$ regulates, $I_2$ does not invest), ($K_1$ adopts, $M_2$ does not regulate, $I_1$ invests), ($K_1$ adopts, $M_2$ does not invest). $I_2$ does not invest), ($K_2$ does not take, $M_1$ regulation, $I_1$ investment), ($K_2$ does not take, $M_1$ regulation, $I_2$ does not invest), ($K_2$ does not take, $M_2$ does not regulate, $I_1$ investment), ($K_2$ does not take, $M_2$ does not regulate, $I_2$ does not invest). According to the parameter assumptions in Table 2, when the strategy combination is ($K_1$ adoption, $M_1$ regulation, $I_1$ investment), the emission control enterprises get higher ESG scores because they buy more forestry carbon sinks, thus obtaining the investment income $E_1$ of the investment institutions and the subsidies $S_1$ of the emission control enterprises which are more active in disclosing ESG information when the government regulates and controls. At the same time, it also needs to pay the cost $C_1$ of purchasing more forestry carbon sinks, and the government needs to pay a certain amount when regulating and controlling.

At the same time, it can obtain the potential benefits $E_2$ brought by the emission control enterprises, but it also needs to pay the subsidies $S_1$ to the enterprises that actively disclose ESG information and the subsidies $S_2$ when the investment institutions implement the investment. Investment institutions need to pay a cost $C_3$ to invest in enterprises with higher ESG ratings, and their investment behavior can get government funding $S_2$ and the potential benefits $E_3$ brought by enterprises purchasing more forestry carbon sinks. Similarly, we can get the returns of emission control enterprises, governments and investment institutions under other strategic combinations.

# 4. Data source and variable selection

In order to verify the mechanism effectiveness of evolutionary game method and explore the influencing factors of enterprises purchasing more forestry carbon sinks, this paper mainly

relies on simulation data generation analysis. By describing the source and generation process of simulation data in detail, and carrying out descriptive statistical analysis of variables, the credibility and scientificity of the study are enhanced.

## 4.1. Data sources

Because this paper is mainly composed of the construction of operation mechanism and the effectiveness of testing mechanism, the data sources are composed of two aspects.

**4.1.1. Reconstruction of ESG rating system and parameter setting of evolutionary game.** According to the characteristics of ESG and carbon market in their respective research fields, this paper adopts Delphi method to reconstruct the ESG rating system of emission control enterprises from three dimensions of scientific research institutions, emission control enterprises and government units.

In order to design the parameters of the evolutionary game model, this paper refers to a large number of field surveys and literature, including government regulation, the behavior of investment institutions and the strategy of emission control enterprises. Through combing the relevant literature, the main cost-benefit parameters of the government, investment institutions and emission control enterprises are set. Based on the field survey and interview analysis, this paper holds that there is a game phenomenon among the government, emission control enterprises and investment institutions, but the game result is not ideal because the ESG system has just started.

**4.1.2. Establishment of indicator system.** Based on the ESG reports of 400 listed companies in high energy-consuming industries and Model weight coefficients of Beijing Forestry University, this paper constructs a general ESG rating system for emission control enterprises according to the common characteristics of the industry, in order to solve the existing problems such as inconsistent ESG rating standards and inconvenient supervision. Based on the complete combing of the ESG rating system of waste water, waste residue and waste gas enterprises, and based on the commonness of heavy pollution industries and the characteristics of the three major industries, the forestry carbon sequestration purchase (GG 5) was innovatively introduced as a secondary index under the primary index greenhouse gas (GG). Finally, the specific evaluation index of environmental dimension under the ESG rating system commonly used by emission control enterprises under the "double carbon" target is integrated, as shown in Table 3.

In order to further design the main parameters of the game subject, according to the interview results and literature, this paper further analyzes the cost and benefit of the three parties according to the strategy choice of the three parties.

(1) The cost and benefit of the government under the regulation and non-regulation strategies. Under the new ESG rating system, when the government chooses to supervise the ESG rating information disclosure of emission control enterprises, the cost and benefit mainly include: the cost of the government to invest a certain amount of manpower and material resources to supervise, the subsidy to the emission control enterprises that actively disclose ESG information, and the fine to the emission control enterprises that do not actively disclose ESG information. Since 2021, the Ministry of Finance and relevant departments have actively supported and fully participated in promoting the formulation and supervision of sustainable disclosure standards. Therefore, the cost of government regulation is assumed to be C2, potential gain E2. At the same time, the subsidy for active disclosure of emission control enterprises is S1, and the fine for non-active disclosure of emission control enterprises is G1. Subsidize S2 for investment institutions that invest according to the ESG scores of emission control enterprises.

**Table 3. Evaluation index of environmental dimension under ESG rating system for heavily polluted enterprises.**

| _ Target Layer | Weight | Criteria Layer (B) | Weight | Indicator Layer (C) | Weight |
|---|---|---|---|---|---|
| Environment (E) A1 | 0.25 | Low Carbon Energy (LC) B11 | 0. 15 | Energy consumption (LC1) C111 | |
| | | | | Energy efficiency (LC2) C112 Energy density (LC3) C113 | |
| | | Water Resources Management (WRM) B12 | 0. 15 | Total Water Consumption (WRM1) C121 Water Intake (WRM2) C122 | 0.40 |
| | | Waste Management (WM) B13 | 0. 15 | Total Waste (WM1) C131 | |
| | | | | Waste output (WM2) C132 Waste disposal volume (WM3) C133 | |
| | | Greenhouse gas (GG) B14 | 0. 25 | $CO_2$ emissions (GG1) C141 $SO_2$ emissions (GG2) C142 $NOx$ emissions (GG3) C143 Soot emissions (GG4) C144 | 0. 15 0. 15 0.20 0.20 |
| | | | | Forestry carbon sequestration purchase (GG5) C145 | 0.30 |
| | | Environmental Investment (EEP) B15 | 0. 35 | Environmental Training and Education (EEP1) C151 | |
| | | | | Environmental protection violation accident (EEP2) C152 | 0.35 |
| | | | | Environmental Early Warning and Emergency Response Mechanism (EEP3) C153 | |

(2) The cost and benefit of controlling emission enterprises in purchasing more forestry carbon sinks and maintaining the existing purchasing strategy. The costs and benefits involved in the purchase of more forestry carbon sinks by emission control enterprises mainly include: the income from the investment of investment institutions, the cost of purchasing more forestry carbon sinks, and the subsidies obtained from the active disclosure of ESG information. The vast majority of listed companies participating in ESG activities bring more benefits than costs, and enterprises increase their value by improving profitability, cash flow or reducing financing costs. Therefore, it is assumed that the cost of enterprise purchase is C1, the investment income is E1, and the government subsidy due to active disclosure of information is S1. At the same time, the potential benefits to investment institutions are E3, and the potential benefits to the government are E2. The costs and benefits involved in maintaining the existing purchase strategy of emission control enterprises mainly include: obtaining the benefits of investment institutions and fines when they do not actively disclose. Therefore, it is assumed that the investment income obtained by maintaining the existing purchasing strategy is E4, and the penalty caused by inactive disclosure is G1. At the same time, the potential return to investment institutions is E5.

(3) The cost and benefit of investment institutions under investment and non-investment strategies. Investment institutions adopt investment strategies based on the ESG scores of emission control enterprises, and the costs and benefits involved mainly include: potential benefits brought by enterprises, government funding, and the cost of investing in enterprises with ESG ratings. Therefore, it is assumed that the potential income of investment institutions from enterprises is E2, the government funding is S2, the cost of investing in enterprises with higher ESG ratings is C3, and the cost of investing in enterprises with lower ESG ratings is C4. At the same time, the investment income brought to the enterprise is E1. The parameters of the above design and their meanings are shown in Table 4.

## 4.2. Descriptive statistics

In order to supplement the empirical analysis, this paper generates data through simulation and makes descriptive statistical analysis of the variables. The descriptive statistical results of

**Table 4. Parameters and their meanings.**

| Parameter | Meaning |
|---|---|
| E1 | Income from investment institutions obtained by enterprises after purchasing more forestry carbon sinks |
| E2 | The potential benefits to the government after enterprises purchase more forestry carbon sinks |
| E3 | Potential benefits to investment institutions after enterprises purchase more forestry carbon sinks |
| E4 | Income from the investment of investment institutions obtained by enterprises maintaining their existing purchasing strategies |
| E5 | The potential benefits of maintaining the existing purchasing strategy for investment institutions |
| C1 | The cost for enterprises to purchase more forestry carbon sinks |
| C2 | The cost of manpower, material resources and financial resources paid by the government when adopting regulatory strategies |
| C3 | The cost of investing in companies with higher ESG ratings |
| C4 | The cost of investing in companies with lower ESG ratings |
| S1 | Subsidies given by the government to enterprises with active ESG information disclosure |
| S2 | Funding from the government received by an investment institution when it makes an investment |
| G1 | Fines imposed on enterprises that are not active in ESG information disclosure during government regulation |

the simulation data variables are shown in Table 5. It is assumed that in the simulation process, enterprises, governments and investment institutions have three different strategies.

# 5. Analysis and simulation of factors affecting forestry carbon sequestration demand

In the process of promoting emission control enterprises to purchase forestry carbon sinks and achieve the goal of "double carbon", it is of great significance to understand the influencing factors of forestry carbon sink demand. This paper will combine empirical analysis and simulation analysis to explore the impact of various factors on forestry carbon sequestration demand, and provide a scientific basis for policy formulation.

## 5.1. Analysis of factors affecting forestry carbon sequestration demand

In order to reveal the driving factors of enterprises' purchase of forestry carbon sequestration, this paper first conducts a benchmark regression analysis to identify the key variables affecting enterprises' purchase decisions. Through the construction of regression model, the influence degree of each factor is quantified, and its role in enterprise decision-making is clarified, and then the heterogeneity analysis of enterprises in different industries is carried out, and the behavior differences of enterprises in different industries in purchasing forestry carbon sequestration are discussed.

**Table 5. Descriptive statistical results of simulation data variables.**

| Variables | Mean value | Standard deviation | Minimum value | Maximum value |
|---|---|---|---|---|
| Forestry carbon sequestration purchased by enterprises | 48.58 | 12.37 | 32. 37 | 72.38 |
| Enterprise ESG score | 78.92 | 8.45 | 58.00 | 88.00 |
| Amount of government subsidies | 23.18 | 8,15 | 12.00 | 34.00 |
| Investment amount of investment institutions | 98.7 | 17.48 | 85.00 | 142.00 |
| Carbon emission reduction/t | 132.48 | 26.87 | 95.00 | 165.00 |
| Forestry carbon sequestration cost/ (yuan/t) | 45.92 | 8.86 | 35.00 | 61.00 |

**Table 6. Benchmark regression results.**

| Variables | OLS model regression result | Complementary log-log model result |
|---|---|---|
| Enterprise ESG score (X1) | 0.4315 (0. 1300) | 0.4182 (0. 1300) |
| Amount of government subsidy (X2) | 0.7218 (0. 1415) | 0.7014 (0. 1576) |
| Investment amount of investment institutions (X3) | 0.37659 (0. 1032) | 0.34814 (0. 1090) |
| Forestry carbon sink cost (X4) | -0.510216(0. 1452) | -0.493087 (0. 1508) |
| Constant term | 11.5432 (2.3500) | 10.8765 (2.3500) |
| Adjust R 2 | 0. 1487 | 0. 1524 |
| Number of samples | 400 | 400 |

Considering that the selected variables may be highly correlated and lead to multicollinearity problems, resulting in bias in coefficient estimation results, this paper uses variance inflation factor (Variance Inflation Factor, VIF) to test the model to determine whether there is multicollinearity problem. Judging from the empirical value, if the VIF is less than 10, it indicates that there is no multicollinearity problem. The test results show that the VIF of all variables is less than 10, so the variables selected in this paper do not have multicollinearity problems.

The influencing factors of forestry carbon sequestration purchased by enterprises are benchmarked and regressed, and the results are shown in Table 6. The ordinary least squares (Ordinary Least Squares, OLS) model regression shows that the ESG score of enterprises, the amount of government subsidies, the investment of investment institutions and the cost of forestry carbon sequestration have a significant impact on the amount of forestry carbon sequestration purchased by enterprises, which proves that the hypothesis H1 is established. At the same time, after using the complementary log-log model to correct the bias, although the estimation coefficient has changed, the standard error of each variable has decreased, and the significance of each variable has not changed basically, that is to say, the marginal effect estimated by the model is basically similar to the regression result of the OLS model, which proves the robustness of the regression result. After using the complementary log-log model to correct the bias, although the estimated coefficients of each variable have changed, their significance has not changed basically, indicating that the regression results are robust. Especially, the standard errors of all variables decreased, which further proved the accuracy of the model estimation. The adjusted R 2 is 0.145, which is close to 0.155 of the OLS model, indicating that the explanatory power of the model is consistent. To sum up, the ESG score of enterprises, the amount of government subsidies, the investment of investment institutions and the cost of forestry carbon sequestration have a significant impact on the amount of forestry carbon sequestration purchased by enterprises. This provides an important reference for policy makers and enterprise managers. It is suggested that when formulating policies, we should consider reducing the cost of forestry carbon sequestration by improving the ESG score of enterprises, increasing government subsidies and investment institutions to promote green investment of enterprises.

Benchmark regression results have confirmed that the ESG score of enterprises, the amount of government subsidies, the amount of investment of investment institutions and the cost of forestry carbon sequestration have a significant impact on the amount of forestry carbon sequestration purchased by enterprises. So, do enterprises in different industries also show consistency characteristics? Are there differences in the degree of influence of these factors? To this end, this paper divides enterprises into nine groups by industry, namely, chemical

**Table 7. Heterogeneity analysis results of different industries.**

| Variables | Chemical Industry | The food industry | Textile industry | Metallurgical industry | Paper Industry | Coal Industry | Thermal Power Industry | Cement industry | Building materials industry |
|---|---|---|---|---|---|---|---|---|---|
| Enterprise ESG score (X1) | 0.4522 | 0.4224 | 0.444 | 0.4152 | 0.4333 | 0.402 | 0.4223 | 0.4142 | 0.432 |
| Amount of government subsidy (X2) | 0.6242 | 0.6052 | 0.6121 | 0.5834 | 0.594 | 0.575 | 0.615 | 0.596 | 0.605 |
| Investment amount of investment institutions (X3) | 0.305 | 0.3162 | 0.3248 | 0.303 | 0.312 | 0.295 | 0.316 | 0.302 | 0.3241 |
| Forestry carbon sink cost (X4) | -0.4781 | -0.45 | -0.471 | -0.431 | -0.451 | -0.42 | -0.461 | -0.439 | -0.449 |
| Constant term | 15. 126 | 14.124 | 14.358 | 13.785 | 14.125 | 13.758 | 14. 752 | 13.814 | 13.652 |
| Adjust R 2 | 0. 1415 | 0. 1575 | 0. 1464 | 0. 1397 | 0. 1454 | 0. 1391 | 0. 1485 | 0. 1429 | 0. 1485 |
| Number of samples | 42 | 113 | 29 | 36 | 22 | 65 | 18 | 27 | 48 |

industry, food industry, textile industry, metallurgy industry, paper industry, coal industry, thermal power industry, cement industry and building materials industry, and conducts heterogeneity analysis on each industry. The results are shown in Table 7, which proves that the hypothesis H2 is established.

(1) The performance of enterprises in different industries is consistent, that is, ESG scores, government subsidies and investment institutions have a significant positive impact on all industries, while the cost of forestry carbon sequestration has a significant negative impact. Although there are some differences in the decision-making of purchasing forestry carbon sequestration among enterprises in various industries, in general, the impact of ESG score, government subsidy amount, investment amount of investment institutions and forestry carbon sequestration cost on purchasing forestry carbon sequestration is consistent in various industries. This shows that improving ESG scores of enterprises, increasing government subsidies and investment institutions, and reducing the cost of forestry carbon sequestration are universally applicable incentives, which can effectively promote green investment of enterprises in different industries.

(2) In different industries, government subsidies play the most significant role in promoting green investment of enterprises. The amount of government subsidies has the greatest impact on the amount of forestry carbon sequestration purchased by enterprises in all industries, indicating that the government's incentive policies play a vital role in various industries.

## 5.2. Analysis of evolution results

Based on the eigenvalue analysis of the Jacobian matrix, eight potential stable points are identified, as shown in Table 8. However, the actual stability status of these points is affected by the specific assumed parameters, that is, whether the constraints are satisfied or not, which requires further verification.

(1) Judge the unstable point. Assuming that $G1 > C2$ has been given, that is, the penalty imposed by the government on the enterprise under the regulation strategy is greater than the cost paid, that is, the realization of "government benefit", then $-C2 + G1 > 0$, so $\lambda2 > 0$ in E1 (0,0,0), then E1, E4, E7 are unstable points; S2-C4 represents the net income when the investment institution invests in an enterprise with a lower ESG rating. Since the investment institution is a rational gambler and the purpose of investment is to make profits, for the investment institution, $S2-C4 > 0$, then $E5-C4 + S2 = (E5-C4) + S2 > 0$. Then E3 is an unstable point;

**Table 8. Stability analysis of equilibrium point.**

| Equilibrium point | Eigenvalue | | | Constraints | Stability |
|---|---|---|---|---|---|
| | $\lambda 1$ | $\lambda 2$ | $\lambda 3$ | | |
| $E1(0, 0, 0)$ | - | + | + | Constraint cannot be satisfied | Unstable |
| $E2(1, 0, 0)$ | + | * | * | Constraint cannot be satisfied | Unstable |
| $E3(0, 1, 0)$ | * | - | + | Constraint cannot be satisfied | Unstable |
| $E4(0, 0, 1)$ | * | + | - | Constraint cannot be satisfied | Unstable |
| $E5(1, 1, 0)$ | * | * | * | $C1—G1—S1 < 0, C2 + S1 < S2, S2—C3 + E3 < 0$ | ESS |
| $E6(1, 0, 1)$ | * | * | * | $E1—E4—C1 > 0, C2 + S1 > 0, S2—C3 + E3 > 0$ | ESS |
| $E7(0, 1, 1)$ | * | + | * | Constraint cannot be satisfied | Unstable |
| $E8(1, 1, 1)$ | * | * | * | Constraint cannot be satisfied | Unstable |

provided that the conditions have given C2 > 0 and S2 > 0, then C2 + S2 > 0, and E8 is an unstable point.

(2) Judge the possible stable point. This article divides into two situations to judge the possible stable point.

Case 1: When C1−G1−S1 < 0, C2 + S1 < S2, S2−C3 + E3 < 0, there exists a stationary point E5 (1, 1, 0) for the replicated dynamical system. In the stable point state, the government has been set to participate in the regulation. G1 + S1 represents the net income when enterprises disclose ESG information actively. At this time, the net income is greater than the cost of purchasing more forestry carbon sinks, indicating that enterprises will choose to purchase more forestry carbon sinks. Contrary to Case 1, the investment institution chooses not to invest, so there is a stable point E5 (1, 1, 0) in the replicated dynamic system. In this case, λ3 > 0 for E6 (1, 0, 1), that is, E6 is an unstable point.

Case 2: When E1−E4−C1 > 0, C2 + S1 > 0, S2−C3 + E3 > 0, the replicated dynamic system has a stable point E6 (1, 0, 1). For enterprises, the profit difference between the two strategies of choosing to buy and not to buy forestry carbon sequestration is greater than 0, that is, the net profit of enterprises choosing to buy forestry carbon sequestration is positive, indicating that enterprises can get more profits by choosing to buy forestry carbon sequestration, so enterprises choose to buy. As far as the government is concerned, the sum of the cost and subsidy of adopting the regulation strategy is positive, so the government will pay more money and choose not to regulate. For investment institutions, S2-C3 + E3 represents the net income of investment institutions to enterprises with higher ratings. When the net income is positive, it indicates that the investment institutions are rational in making the investment decision, so they will choose to invest. At this time, E6 is a stable point. In this case, λ3 > 0 for E5 (1, 1, 0), and E5 is an unstable point.

## 5.3. Model validation

The model used in this article belongs to the classic model in evolutionary game theory and has been extensively used by researchers [46–51]. To further explore this issue, this paper utilized Chen's data and conducted numerical simulation analysis. MATLAB was used to perform the Pareto optimal state combination mentioned above to verify the effectiveness of evolutionary game stability analysis and the sensitivity of each agent to parameters [52].

We shift our focus to the buyers of forestry carbon sinks:Coal Power Enterprises (CPEs) and New Energy Power Enterprises(NEPEs).According to Chen's calculation simulation: Under the condition of satisfying the parameter setting conditions of Pareto optimality, this article considers the actual situation and relevant expert opinions compre hensively, and the

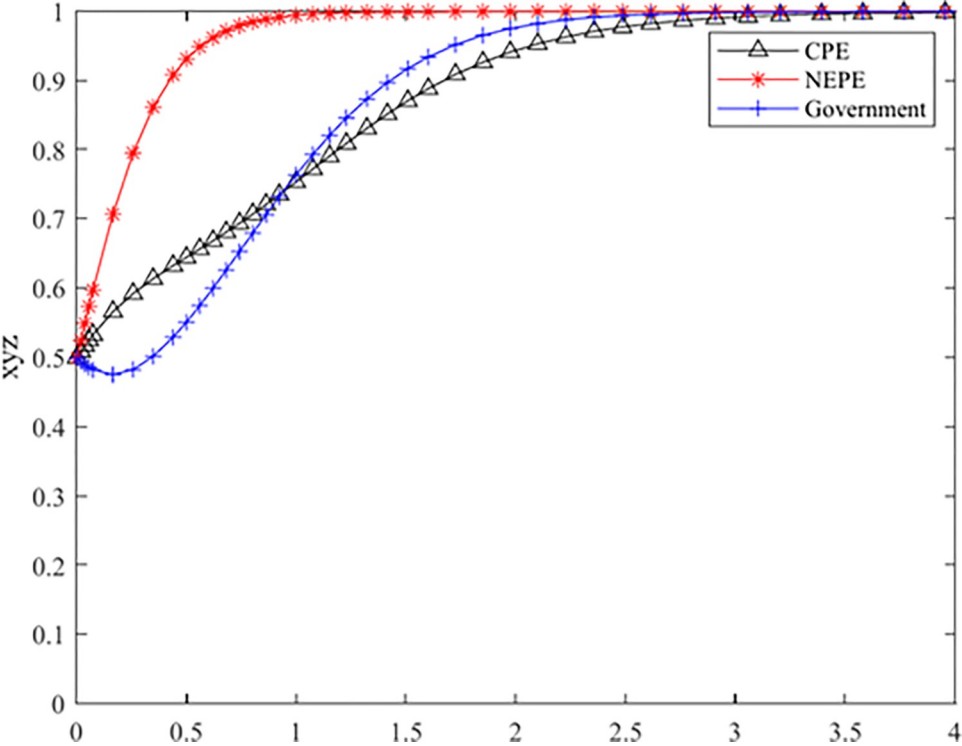

**Fig 4. Parameter evolution simulation results.**

initial values of the parameters are as follows: R1 = 2, R2 = 4, Rs = 3, C1 = 2, C2 = 1, Cg = 4, A1 = 3, A2 = 2, B1 = 3, B2 = 3, F1 = 2, F2 = 3. The above parameters are brought into the tripartite game evolution system, and numerical simulation analysis is conducted with MATLAB R2021a software. The results are shown in Fig 4.

Fig 4 illustrates the evolutionary trend in the tripartite game under joint operation, where it can be seen that the initial NEPE had a stronger inclination to choose joint operation due to the low-cost supervision value of the CPE. The government initially preferred a loose supervision strategy, and as the other two entities' willingness to choose joint operation increased, the government's willingness to choose a strict supervision strategy is also increased. When the NEPE evolved to joint operation, the government's rate of choosing strict supervision was faster than that of CPE choosing joint operation. Fig 3 shows that the system ultimately tends towards the ideal state of (1, 1, 1), where the CPE chooses joint operation, the NEPE chooses joint operation, and the government implements strict supervision, indicating the validity of the conclusion.

Fig 5 shows that with the increase in additional operating income that the CPE can obtain through joint operation, its willingness to choose joint operation becomes stronger, and government is more willing to choose strict supervision.

Fig 6 shows that with the increase in additional operating income that the NEPE can obtain through joint operation, its willingness to choose joint operation becomes stronger, and the government has a greater willingness to choose strict supervision. These simulation conclusions are consistent with our model calculation conclusions, proving the effectiveness and reliability of the model

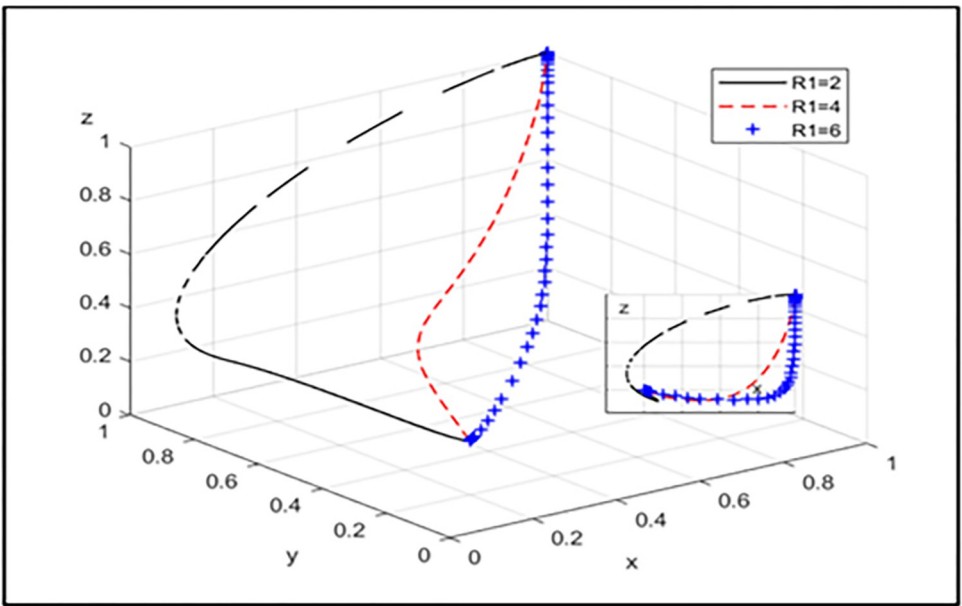

**Fig 5. The influence of the additional operating income of the CPE choosing joint operation.**

## 6. Conclusions

This paper combines Guizhou's carbon market and corporate environmental information disclosure, two macro and micro levels of emission control methods, through the construction of ESG rating system of emission control enterprises, activates the forestry carbon sequestration market with the help of ESG function, and uses evolutionary game model to verify the

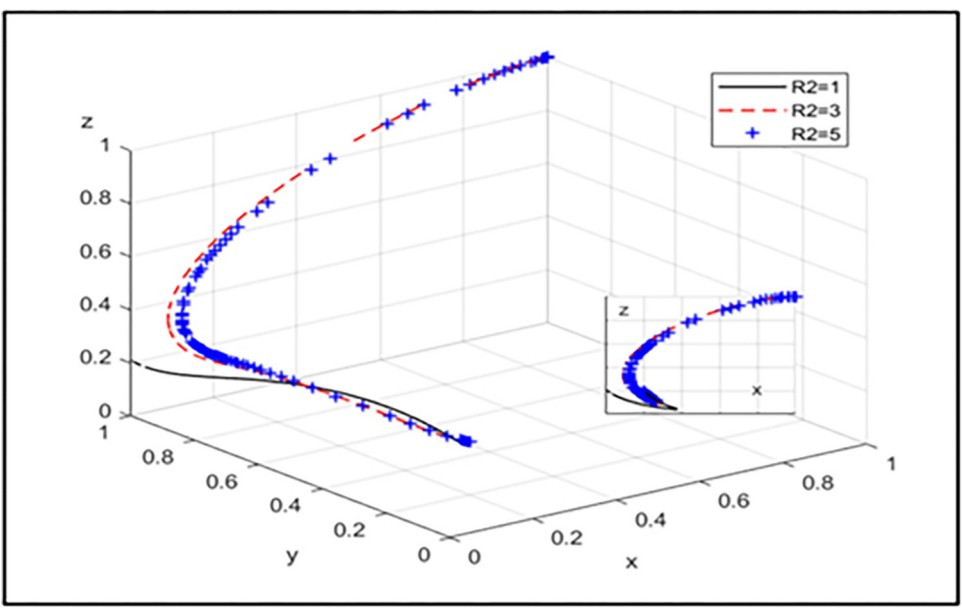

**Fig 6. The influence of the additional operating income of the NEPE choosing joint operation.**

effectiveness of this mechanism, draws research conclusions and discusses them, and finally puts forward policy implications.

## 6.1 Research conclusion

Based on the analysis of influencing factors of forestry carbon sequestration demand and evolutionary game simulation, this paper discusses the influence of various factors on the purchase behavior of forestry carbon sequestration from both theoretical and empirical perspectives. At the same time, the simulation results of the evolutionary game model further verify the interaction and influence mechanism of these factors in the actual situation, and four conclusions are drawn.

(1) The regression results show that the ESG score of enterprises and the amount of government subsidies have a significant positive impact on the amount of forestry carbon sequestration purchased by enterprises, while the cost of forestry carbon sequestration has a significant negative impact on it, and there is heterogeneity in different industries. It shows that improving ESG score, increasing government subsidies and reducing the purchase cost of forestry carbon sequestration are important ways to promote green investment of enterprises. In addition, the heterogeneity analysis of different industries shows that although the direction of the influencing factors is consistent, the degree of influence is different, especially the role of government subsidies in the chemical, food and textile industries is particularly prominent. It can be seen that different industries have specific sensitivities in responding to policy incentives and need to formulate policies according to local conditions.

(2) The stable points of the evolutionary game are E5 (1,1,0) and E6 (1,0,1), which shows that the ESG rating system after the introduction of forestry carbon sequestration can spontaneously promote the purchase of forestry carbon sequestration by emission control enterprises from the market mechanism, and verifies the effectiveness of the operation mechanism of forestry carbon sequestration activating the ESG function of emission control enterprises. From the results of stability analysis, it can be seen that the conclusion is robust.

(3) The higher the cost of purchasing forestry carbon sequestration, the lower the purchasing willingness of controlling emission enterprises, and the higher the investment willingness of investment institutions. Therefore, if the government wants to improve the probability of enterprises purchasing forestry carbon sequestration and promote the investment enthusiasm of investment institutions, it can consider setting a ladder price of forestry carbon sequestration to control the purchase cost of forestry carbon sequestration buyers at a relatively balanced level, so as to maintain the high investment enthusiasm of investment institutions and promote enterprises to purchase forestry carbon sequestration.

(4) The higher the government's subsidies to enterprises that actively disclose ESG information, the higher the purchase willingness of emission control enterprises, while the investment willingness of investors decreases. Therefore, in order to balance the interests of both enterprises and investment institutions, promote enterprises to increase purchases and stimulate investment enthusiasm, the government should set a reasonable level of subsidies, because excessive subsidies to enterprises that actively disclose ESG information may weaken the enthusiasm of enterprises for financing, or lack of motivation for improvement, which will increase the investment risk of investment institutions, thus inhibiting investment behavior.

## 6.2 Policy implications

At present, emission control enterprises in the carbon market mainly solve the problem of excess emissions by purchasing excess carbon quotas, but this way can not really offset excess carbon emissions from the ecological level. Therefore, promoting the activity of forestry

carbon sequestration market is of great significance to the process of carbon neutralization. This paper introduces the forestry carbon sequestration index into the ESG rating system, and verifies the effectiveness of the operating mechanism.Combined with the research findings, this paper puts forward five policy implications.

(1) Expedite the development of a localized and standardized ESG rating framework specifically for enterprises involved in emission control. The current domestic ESG rating system lacks uniformity, resulting in significant discrepancies in the information disclosed across various industries. For instance, within the high energy-consuming sector, there are distinct categories such as wastewater, waste residue, and waste gas industries. The ESG reports produced by leading publicly listed companies in these sectors tend to be heavily focused on specific aspects. Therefore, it is imperative to accelerate the standardization of the ESG framework to achieve a truly quantifiable rating system. Concurrently, in alignment with China's policy objectives and regional strengths, it is essential to enhance the existing ESG framework. This should not involve merely replicating international ESG standards; rather, it should focus on localized innovations. For example, incorporating indicators for forestry carbon sequestration can leverage China's abundant forest resources and facilitate the achievement of the nation's "dual carbon" goals.

(2) For control and discharge enterprises, improve the ESG disclosure system, strengthen the supervision of the quality of ESG information disclosure, and avoid the chaos of uneven quality of ESG information; At the same time, subsidies will be given to the emission control enterprises that actively disclose ESG information, and administrative penalties will be imposed on the emission control enterprises that improperly verify ESG information. For investment institutions, a floating incentive mechanism should be set up to strengthen the supervision of ESG rating system, enhance the confidence of investment institutions in making investment decisions based on ESG rating indicators, reduce the risk of long-term sustainable investment by investment institutions, make investment institutions feel sustainable benefits, and ultimately form a virtuous circle [53–56].

(3) It is suggested that the government should continue to increase subsidies for high-carbon enterprises, especially in high-pollution industries, and encourage enterprises to purchase forestry carbon sinks through fiscal and taxation policies to promote green transformation. At the same time, enterprises should pay attention to improving their ESG scores, not only to obtain more investment support, but also to enhance market competitiveness. In view of the high purchase cost of forestry carbon sequestration, it is suggested that the government and relevant institutions should take measures to reduce the investment cost of enterprises, such as optimizing forestry carbon sequestration trading through technological innovation and market mechanism, and improving the enthusiasm of enterprises to participate in it, so as to achieve the goal of sustainable development.

(4) Relax the current standards of CCER methodology and reasonably reduce the entry threshold of forestry carbon sequestration projects. The current CCER methodology is still relatively strict, and the forests that meet the methodology are limited, which limits the supply of forestry carbon sequestration projects and restricts the development of forestry carbon sequestration market. Since the restart of CCER in October 2023, the forestry carbon sequestration projects developed before 2017 can be re-traded in the carbon sequestration market, but the declaration channel of the newly developed forestry carbon sequestration projects has not yet been opened, so it is suggested to open a new declaration channel as soon as possible to increase the supply of CCER forestry carbon sequestration projects [57,58].

(5) Optimize and design the quota of emission control enterprises. According to the actual survey, the carbon quota in the carbon sink market is generally sufficient, resulting in a lower price than CCER, which has a natural price advantage. In order to save costs, emission control

enterprises usually choose to buy carbon quotas to offset their excess carbon emissions. In order to make emission control enterprises spontaneously choose to purchase forestry carbon sinks to reduce emissions, in addition to using the investment impact mechanism of ESG rating system, we should also enhance the relative price advantage of forestry carbon sinks purchase from the market perspective. The government should optimize the design of quotas, reduce the surplus carbon quotas in the market, gradually control the carbon quotas in the market in the process of promoting the green transformation of emission control enterprises, not too loose or too tight at once, and scientifically promote the realization of the "double carbon" goal on the premise of ensuring the normal development of the economy.

(6) The forestry carbon market in Guizhou Province is currently not sound, and the forestry carbon trading platform is not yet complete. This greatly inhibits the circulation of forestry carbon sinks in Guizhou Province, reduces the economic benefits of forestry carbon sinks, increases the information search cost of forestry carbon sink project operators, restricts the willingness and enthusiasm of forestry operators to provide forestry carbon sinks, and suppresses the construction of forestry carbon sink benefit sharing bodies. Therefore, government agencies should. Firstly, the forestry carbon sink market should be constructed in stages. According to its current development status, the development stages of the forestry carbon sink market in Guizhou Province can be divided into the basic construction stage, the experimental operation stage, and the development and improvement stage. In the infrastructure construction stage, the government should establish a reasonable and specific operating mechanism for the forestry carbon market, build a framework for the operation of the forestry carbon market, establish a sound forestry carbon trading platform and information query system, and lay the foundation for the operation of the forestry carbon market. During the trial operation phase, government agencies should focus on improving the circulation of forestry carbon sinks within the province, actively promoting communication and interaction between the supply and demand sides of carbon sinks through exhibitions and promotional events, improving the trading efficiency of forestry carbon sinks, and promoting the effective realization of the economic benefits of forestry carbon sinks. Secondly, promote the transformation of the driving force of the forestry carbon sink market in Guizhou Province. In the infrastructure construction and experimental operation stages of the forestry carbon sequestration market in Guizhou Province, policies are its main driving force. The government ensures the "commodification" of forestry carbon sinks and the basic operation of the forestry carbon sink market by formulating relevant policies. With the orderly development of the forestry carbon market, the regulatory and resource allocation capabilities of the forestry carbon market will continue to improve. In this context, in order to maximize the market functions of resource allocation, price discovery, and supply-demand balance in the forestry carbon sequestration market, the driving force of the market should gradually shift from policy led to market led, policy assisted, and ultimately fully market driven, in order to further promote the development of forestry carbon sequestration in Guizhou Province.

## Supporting information

**S1 File. Part of the data in this article comes from the "Guizhou Forestry Yearbook" section.** I have downloaded the complete yearbook report and uploaded it as an attachment for data support.
(DOCX)

## Author Contributions

**Data curation:** Cui Tao.

**Formal analysis:** Cui Tao.

**Investigation:** Yan Jun.

**Methodology:** Yan Jun.

**Writing – original draft:** Wu Yang.

**Writing – review & editing:** Zhang Min.

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
