## [Decision Letter · Decision Letter 0]

23 Sep 2024

PONE-D-24-34694Exploration of Forestry Carbon Sequestration Practice Path in Guizhou Province-Based on Evolutionary Game ModelPLOS ONE

Dear Dr. Yang,

Thank you for submitting your manuscript to PLOS ONE. After careful consideration, we feel that it has merit but does not fully meet PLOS ONE’s publication criteria as it currently stands. Therefore, we invite you to submit a revised version of the manuscript that addresses the points raised during the review process.<ol><li>**Clarity and Methodological Detail**:

Multiple reviewers pointed out a lack of clarity in data presentation and methodology. Specific details about data sources, parameter assumptions, and simulation processes need to be elaborated.<li>**Theoretical Framework and Literature Review**:

The literature review requires a more critical analysis and discussion of gaps in existing research. The theoretical framework also needs to be strengthened with better justification for the chosen model.<li>**Generalizability and Limitations**:

The study’s geographic limitation to Guizhou Province is a concern. The authors should discuss how findings may vary in different contexts and explore scenarios under varying conditions.Limitations of the evolutionary game model, such as assumptions of rational behavior, should be explicitly stated to provide a balanced view of the findings.<li>**Robustness and Sensitivity Analysis**:

Reviewers suggested conducting sensitivity analyses on key parameters to test the robustness of results, which is critical for validating findings.<li>**Comprehensive Discussion and Policy Implications**:

The policy recommendations are considered strong, but more depth is needed regarding potential challenges in implementation and the broader implications of the findings.<li>**Language and Structural Issues**:

Several grammatical errors and structural issues were noted. A thorough proofreading and refinement of the manuscript’s organization would enhance readability.

We look forward to receiving your revised manuscript.

Kind regards,

Pradeep Paraman

Academic Editor

PLOS ONE

 1. When submitting your revision, we need you to address these additional requirements. Please ensure that your manuscript meets PLOS ONE's style requirements, including those for file naming. The PLOS ONE style templates can be found at https://journals.plos.org/plosone/s/file?id=wjVg/PLOSOne_formatting_sample_main_body.pdf and https://journals.plos.org/plosone/s/file?id=ba62/PLOSOne_formatting_sample_title_authors_affiliations.pdf 2. Please provide additional details regarding participant consent. In the ethics statement in the Methods and online submission information, please ensure that you have specified (1) whether consent was informed and (2) what type you obtained (for instance, written or verbal, and if verbal, how it was documented and witnessed). If your study included minors, state whether you obtained consent from parents or guardians. If the need for consent was waived by the ethics committee, please include this information. If you are reporting a retrospective study of medical records or archived samples, please ensure that you have discussed whether all data were fully anonymized before you accessed them and/or whether the IRB or ethics committee waived the requirement for informed consent. If patients provided informed written consent to have data from their medical records used in research, please include this information. 3. You indicated that ethical approval was not necessary for your study. We understand that the framework for ethical oversight requirements for studies of this type may differ depending on the setting and we would appreciate some further clarification regarding your research. Could you please provide further details on why your study is exempt from the need for approval and confirmation from your institutional review board or research ethics committee (e.g., in the form of a letter or email correspondence) that ethics review was not necessary for this study? Please include a copy of the correspondence as an ""Other"" file. 4. We note that the grant information you provided in the ‘Funding Information’ and ‘Financial Disclosure’ sections do not match.  When you resubmit, please ensure that you provide the correct grant numbers for the awards you received for your study in the ‘Funding Information’ section. 5. Thank you for stating the following financial disclosure: "This research was supported by Regional Project of National Natural Science Foundation of China (41463003), Surface Project (41573043), Concealed Ore Deposit Exploration and Innovation Team of Guizhou Colleges and Universities (Guizhou Education and Cooperation Talent Team [2015]56), Provincial Key Discipline of Geological Resources and Geological Engineering of Guizhou Province (ZDXK[2018]001), Huang Danian Resources of National colleges and universities Teachers' Team of Exploration Engineering (Teacher Letter [2018] No. 1), Geological Resources and Geological Engineering Talent Base of Guizhou Province (RCJD2018-3), Key Laboratory of Karst Engineering Geology and Hidden Mineral Resources of Guizhou Province (Qianjiaohe KY [2018] No. 486Guizhou Institute of Technology Rural Revitalization Soft Science Project(2022xczx10), Education and Teaching Reform Research Project of Guizhou Institute of Technology (JGZD202107,2022TDFJG01)." Please state what role the funders took in the study.  If the funders had no role, please state: "The funders had no role in study design, data collection and analysis, decision to publish, or preparation of the manuscript." If this statement is not correct you must amend it as needed. Please include this amended Role of Funder statement in your cover letter; we will change the online submission form on your behalf. 6. We note that your Data Availability Statement is currently as follows: "All relevant data are within the manuscript and its Supporting Information files." Please confirm at this time whether or not your submission contains all raw data required to replicate the results of your study. Authors must share the “minimal data set” for their submission. PLOS defines the minimal data set to consist of the data required to replicate all study findings reported in the article, as well as related metadata and methods (https://journals.plos.org/plosone/s/data-availability#loc-minimal-data-set-definition). For example, authors should submit the following data: - The values behind the means, standard deviations and other measures reported;- The values used to build graphs;- The points extracted from images for analysis. Authors do not need to submit their entire data set if only a portion of the data was used in the reported study. If your submission does not contain these data, please either upload them as Supporting Information files or deposit them to a stable, public repository and provide us with the relevant URLs, DOIs, or accession numbers. For a list of recommended repositories, please see https://journals.plos.org/plosone/s/recommended-repositories. If there are ethical or legal restrictions on sharing a de-identified data set, please explain them in detail (e.g., data contain potentially sensitive information, data are owned by a third-party organization, etc.) and who has imposed them (e.g., an ethics committee). Please also provide contact information for a data access committee, ethics committee, or other institutional body to which data requests may be sent. If data are owned by a third party, please indicate how others may request data access.

Reviewers' comments:

Reviewer's Responses to Questions

**Comments to the Author**

1. Is the manuscript technically sound, and do the data support the conclusions?

Reviewer #1: Yes

Reviewer #2: Yes

Reviewer #3: Yes

Reviewer #4: Partly

Reviewer #5: Yes

Reviewer #6: Yes

Reviewer #7: Yes

2. Has the statistical analysis been performed appropriately and rigorously? 

Reviewer #1: Yes

Reviewer #2: Yes

Reviewer #3: Yes

Reviewer #4: Yes

Reviewer #5: Yes

Reviewer #6: Yes

Reviewer #7: I Don't Know

3. Have the authors made all data underlying the findings in their manuscript fully available?

Reviewer #1: No

Reviewer #2: Yes

Reviewer #3: Yes

Reviewer #4: No

Reviewer #5: Yes

Reviewer #6: No

Reviewer #7: Yes

4. Is the manuscript presented in an intelligible fashion and written in standard English?

Reviewer #1: Yes

Reviewer #2: Yes

Reviewer #3: Yes

Reviewer #4: Yes

Reviewer #5: Yes

Reviewer #6: Yes

Reviewer #7: Yes

5. Review Comments to the Author

Reviewer #1: Dear authors

In the study “Exploration of Forestry Carbon Sequestration Practice Path in Guizhou Province-Based on Evolutionary Game Model’’. This study is interesting and important for addressing the environmental pollution in recent decades and increased the forestry carbon sequestration purchased by enterprises for carbon emission reduction. The study evaluates the impact of ESG score of enterprises through regression analysis. The methodology must be clearer in the abstract section. The following major issues must be considered carefully.

1- The novelty and methodology, must be presented carefully and clearly in the abstract

2-The abstract section needs to be clearer and more effective.‎ It is suggested to be re-written summarizing a short introduction, problem statement, ‎methodology, major results, and conclusion and recommendation

3- Implication of study results, conclusion, and recommendation for future must be well defined in the abstract

4- The abbreviations as such CCER and ESG must put in full name in in the abstract and throughout the manuscript

4- Introduction: Carbon enterprises, needs to be introduced in more detail, as its types, distribution in China and globally and impacts on environmental and public health over the last years

5- What are the strategies and technologies used to control the Carbon enterprises?

6- The advantages and disadvantages of building of carbon market over recent years

What criteria were used to select the control technologies?

7- How to validity of the evolutionary game model? How was the data from different sources synthesized and analyzed to ensure consistency and reliability?

8-What are the key gaps in the current understanding of ESG impact on socioeconomic and ecosystems? How do these gaps inform future research directions and policy development?

8- The study identifies several areas for future research, such as the need for more stringent monitoring and tailored source control strategies.

What specific methodologies or technologies hold the most promise for advancing the understanding and management of ESG ? How can these be incorporated into future research efforts?

9- How does this study build upon or differ from previous studies on their impacts? What new insights or perspectives does it offer?

10.What criteria were used to select the studies included in this model? How were these criteria applied to ensure the inclusion of relevant and high-quality application?

11.How was the data from this model synthesized and analyzed? What methods were used to ensure consistency and reliability in the findings?

12.What are the key gaps in the current study? How do these gaps inform the recommendations for future research and policy development?

13.How could the structure to be improved to enhance readability and accessibility for a broader audience?

14- More figures and tables must be added to strengthen and eases the article contents by the readers

15- Limitations of this study must state clearly

16- Conclusion must be customized stating the main results and the significant impact of this study on economics, society and the recommendation for future

17- The reference. The author needs to provide more updated references, especially in the introduction and discussion.

Best regards

Reviewer #2: 1. Generally, it is better to integrate the section “2.1. Status quo of ESG rating system” into the introduction section and make it more concise. The current structure reduces the flow of logic and puts too much efforts in the background rather than research methods and findings.

2. More background should be provided regarding to the evolutionary game model. For example, why used this model and not the other models to do the analysis should be included in the introduction section.

3. Is it proper to assume and conclude the positive impact of ESG scores on the number of forestry carbon sinks purchased by enterprises? I agree with the point that a higher score will result in higher purchases. But the purchase amount itself will also increase the score. The “positive relationship” might be more accurate. Or change “amount of forestry carbon sinks purchased by enterprises” to “the potential forestry carbon sink acquisitions by enterprises”.

4. Lack of references. For example, this research proposed a new application of the model. But the references for the model and its relevant parameters were not listed.

5. The article needs to be critically re-proofread. A few mistakes existed.

e.g.,

1) Titles of section 4.1.1 and 4.1.2 are the same.

2) Table 3: there is an “_” in front of the header.

3) Line 510: “andModel”.

4) Please double check the order of first and last names for all authors. I noticed there are some publications with different orders.

Reviewer #3: This research paper explores factors influencing forestry carbon sequestration demand by emission control enterprises in China, using an evolutionary game model approach. The study makes several valuable contributions: 1) It incorporates forestry carbon sequestration indicators into the ESG rating system for emission control enterprises, providing an innovative mechanism to promote green investment. 2) Through regression analysis and simulation, it identifies key factors affecting enterprises' forestry carbon sink purchases, including ESG scores, government subsidies, and carbon sink costs. 3) The evolutionary game model verifies the effectiveness of the proposed mechanism in promoting voluntary carbon sink purchases by enterprises. 4) The research offers policy recommendations for standardizing ESG ratings, optimizing government subsidies, and improving the carbon trading market. While the methodology is sound and the findings are insightful, the paper could benefit from a more detailed discussion of limitations and future research directions. Overall, this study provides valuable insights for policymakers and researchers in the field of carbon neutrality and sustainable finance.

1. The introduction provides a good overview of the research context, but it could benefit from a more explicit statement of the research objectives and hypotheses. How do the authors expect the incorporation of forestry carbon sequestration indicators into the ESG rating system to specifically impact enterprise behavior?

2. In the literature review section, the authors mention several studies on ESG rating systems and their impact on corporate behavior. Could they provide a more critical analysis of these studies, highlighting any gaps or contradictions in the existing literature that this research aims to address?

3. The methodology section describes the use of the Analytic Hierarchy Process (AHP) for determining index weights. Could the authors provide more details on how the expert group for the Delphi method was selected and what criteria were used to ensure a diverse and representative panel of experts?

4. The paper mentions that "Based on the ESG reports of 400 listed companies in high energy-consuming industries" were used to construct the ESG rating system. How were these companies selected? Were there any specific criteria for inclusion or exclusion? This information would help readers understand the representativeness of the sample.

5. In Table 3, the authors present the evaluation index of the environmental dimension under the ESG rating system. Could they elaborate on why these specific indicators were chosen and how they compare to international ESG standards? This would help readers understand the rationale behind the localized ESG system.

6. The simulation data generation process is briefly mentioned but not fully explained. Could the authors provide more details on how the simulation data was generated and validated to ensure its reliability and representativeness of real-world scenarios?

7. In the regression analysis, the authors find that ESG scores, government subsidies, and carbon sink costs significantly impact forestry carbon sequestration purchases. Have they considered potential interaction effects between these variables? For example, does the impact of ESG scores vary depending on the level of government subsidies?

8. The evolutionary game model is a key component of this research. Could the authors provide a more detailed explanation of why this specific model was chosen over other potential methods for analyzing the interaction between enterprises, governments, and investment institutions?

9. In the results section, the authors state that "The stable points of the evolutionary game are E5 (1,1,0) and E6 (1,0,1)." Could they provide a more detailed interpretation of what these stable points mean in practical terms for the behavior of enterprises, governments, and investment institutions?

10. The paper discusses heterogeneity in different industries regarding the impact of various factors on forestry carbon sequestration purchases. Have the authors considered conducting a more in-depth analysis of why these differences exist and what implications they might have for industry-specific policies?

11. In the conclusion, the authors state that "the ESG rating system after the introduction of forestry carbon sequestration can spontaneously promote the purchase of forestry carbon sequestration by emission control enterprises from the market mechanism." Could they provide more evidence or explanation to support this claim?

12. The policy implications section provides several recommendations. Could the authors discuss potential challenges or barriers to implementing these recommendations, particularly in the context of China's current regulatory environment?

13. The study could be improved by integrating insights from recent research in related fields: https://doi.org/10.3390/ma15207098; https://doi.org/10.1016/j.conbuildmat.2023.132604; https://doi.org/10.1016/j.clema.2022.100111

14. The study focuses on the environmental dimension of ESG ratings. Have the authors considered how the social and governance dimensions might interact with or influence the environmental aspects, particularly in relation to forestry carbon sequestration?

15. While the paper provides valuable insights, it would benefit from a more explicit discussion of its limitations and potential areas for future research. For example, how might the findings be affected by changes in carbon pricing or technological advancements in carbon capture?

Reviewer #4: General Assessment:

• Technical Rigour: The study presents a clear, well-structured approach to evaluating the forestry carbon sequestration practice path using the evolutionary game model. However, there are areas where clarification or expansion of specific sections would enhance the manuscript's rigor.

• Clarity and Coherence: While the manuscript is generally well-written, there are several areas where language improvements or clarifications are necessary. These would improve readability for non-specialists and ensure better communication of key points.

Specific Comments:

Abstract

1. Page 1, Lines 12-26:

o Clarity: The abstract is somewhat dense and could benefit from clearer language. For instance, the sentence, "the ESG score and the amount of government subsidies have a significant positive impact on the amount of forestry carbon sequestration purchased by enterprises..." could be simplified for a broader audience.

o Recommendation: Consider breaking down complex ideas into simpler statements and possibly reducing jargon.

Introduction

2. Page 2, Lines 33-91:

o Context: The introduction effectively outlines the importance of China's "double carbon" goals and sets the stage for the research. However, there is a lack of international context. How do China's efforts compare to other countries?

o Recommendation: Briefly mention global efforts in forestry carbon sequestration to provide international context.

3. Page 2, Line 86-91:

o Linkage: While the introduction covers the macro and micro aspects of carbon sequestration and ESG, the linkage between these two levels can be made clearer.

o Recommendation: Explicitly outline how the macro and micro designs combine to create the proposed mechanism.

Literature Review

4. Page 3-4, Lines 92-205:

o Critical Review: The literature review is comprehensive, but it could benefit from more critical analysis of the gaps in previous research, particularly regarding the shortcomings of ESG systems globally.

o Recommendation: Strengthen the critique of past research to better justify the need for the current study.

Methodology

5. Page 6, Lines 207-248:

o Game Model Clarification: The operation mechanism and evolutionary game model are central to the paper, but the explanation is overly complex and technical. Simplification and additional explanation would benefit readers unfamiliar with game theory.

o Recommendation: Provide a short primer on evolutionary game theory to help readers who may not be familiar with it. Adding a diagram could also aid understanding.

6. Page 9, Line 407:

o Weighting Criteria: The paper uses the Analytic Hierarchy Process (AHP) to determine weights in the ESG rating system. The AHP process is mentioned, but further details on how experts were chosen and how scores were aggregated are needed.

o Recommendation: Provide more detail on the AHP process, including how experts were selected and how subjectivity was minimized.

Results

7. Page 14, Lines 572-602:

o Descriptive Statistics: The results section is clear, but the descriptive statistics could benefit from more discussion. How do these statistics compare with existing data or studies in this field?

o Recommendation: Compare the descriptive statistics with similar studies to highlight the uniqueness of your findings.

Discussion

8. Page 15, Lines 626-655:

o Policy Implications: The discussion mentions policy implications, but they are somewhat general.

o Recommendation: Be more specific about the implications of your findings for policymakers, particularly in terms of forestry carbon sequestration policies in China and abroad.

9. Page 15-16, Lines 633-641:

o Industry Comparison: The heterogeneity analysis across industries is a strong point of the paper. However, more detail is needed on why certain industries (e.g., chemical or textile industries) are affected differently.

o Recommendation: Expand on the reasons behind the varying effects of ESG and government subsidies across different industries.

Writing and Organization

10. General Writing:

o Language: There are several instances where the language is overly technical or complex.

o Recommendation: Simplify language and sentence structure throughout the manuscript for broader accessibility.

11. Structure: The manuscript is well-structured but would benefit from clearer section transitions.

o Recommendation: Use more explicit headings or sub-headings to break up dense sections, particularly in the methodology and results.

In general, the study provides important insights into forestry carbon sequestration in China and offers a novel approach using an evolutionary game model. However, improvements can be made in terms of clarity, depth of analysis, and the presentation of results.

Reviewer #5: The behavior decision-making of environmental dimension of emission control enterprises is significantly constrained with the rise of ESG rating system in China. This paper discusses the impact of ESG score of enterprises through regression analysis, the amount of government subsidies and the cost of forestry carbon sequestration on the purchase of forestry carbon sequestration. The paper requires some correction.

1. Abbreviations are used in the abstract or at most places of the paper. Interpret for the first time and then use abbreviations e.g. ESG, CCER and etc.

2. In the abstract section, the need and scope of the study should be included. At the end of the abstract include some quantitative results

3. Check the Keywords. What do you mean by Keywords ESG.

4. The objectives and research insight questions must be stated at the last paragraph of introduction section.

5. What are specific values in the evolutionary game parameter assumptions and what is the criteria to choose values of these parameters.

6. How are the specified values in Table 3 obtained? Do these values have significant physical meaning?

Reviewer #6: I have thoroughly reviewed the content of the manuscript and would like to provide the following comments: "Exploration of Forestry Carbon Sequestration Practice Path in Guizhou Province Based on Evolutionary Game Model".

1. The research addresses a relevant and timely topic of promoting forestry carbon sequestration to achieve carbon neutrality goals in China. Exploring mechanisms to incentivize emission control enterprises to purchase forestry carbon sinks is of great practical and policy significance.

2. Incorporating forestry carbon sequestration indicators into the ESG rating system for emission control enterprises is an innovative approach. It effectively links macro carbon market mechanisms with micro-level corporate environmental performance. Testing this operational mechanism through an evolutionary game model provides useful theoretical and empirical insights.

3. The manuscript is well-structured and the research framework is clearly presented. The authors systematically analyze the operation mechanism, construct research hypotheses, determine ESG index weights using AHP, and then test the effectiveness of the mechanism through an evolutionary game simulation. The methodology is sound.

4. The findings regarding the positive impact of ESG scores, government subsidies, and investment amounts as well as the negative impact of forestry carbon sequestration costs on enterprises' purchasing behavior provide actionable policy implications. The heterogeneity analysis across different industries further enhances the practical relevance.

5. The conclusions and policy recommendations, such as accelerating ESG rating system construction, strengthening ESG information disclosure supervision, optimizing carbon quotas, etc. are insightful and can guide policymakers in promoting forestry carbon sequestration markets.

Suggestions for Improvement:

1. While the manuscript mentions field surveys and interviews, more details can be provided on the data collection process, sample characteristics, questionnaire design, etc. to enhance the credibility of the qualitative inputs used in the evolutionary game parameter setting.

2. The description of the simulation data generation process for testing influencing factors can be further elaborated. Providing the codes/equations used to generate the simulation data in an appendix would allow replicability.

3. Perform sensitivity analysis on key parameters of the evolutionary game model to test the robustness of the findings. Discuss how changes in parameters like government subsidies, carbon sequestration costs, etc. impact the equilibrium outcomes.

4. Discuss the limitations of the study more explicitly. For example, the evolutionary game assumes rational behavior but bounded rationality of actors can influence outcomes. The simulation is a simplified representation of the real dynamics. Highlighting such limitations helps interpret the findings appropriately.

5. Proofread the manuscript thoroughly to fix minor language and grammatical errors. Ensure consistency in abbreviations, figure and table numbers, references, etc.

Overall, this is a well-executed research that makes valuable contributions to the field of forestry carbon sequestration and ESG integration. With some minor revisions to enhance clarity and robustness, it has the potential to be a strong publication. I commend the authors for their systematic approach and relevant policy recommendations. I hope these comments are helpful in further refining this interesting work.

Title and Abstract:

- The abstract (lines 12-30) provides a good overview, but a key detail is missing. For example, mention the specific research methods used (e.g., AHP, evolutionary game model).

Introduction (Section 1):

- The background and significance of the research are well-established. However, the problem statement could be more focused. Clearly, briefly and concisely state the research gap and the specific objectives of this study (lines 33-91).

- The literature review (Section 2, lines 92-205) covers relevant studies but comes low on critical analysis. Discuss the limitations of existing research more explicitly to highlight the value-addition of this study.

Theoretical Analysis Framework and Research Methods (Section 3):

- The theoretical framework (Section 3.1, lines 207-326) is logically structured. However, the description of the evolutionary game mechanism (lines 263-326) is quite lengthy. Consider presenting the key aspects more concisely.

- The research hypotheses (Section 3.2, lines 327-400) are clearly stated. But the rationale behind each hypothesis could be strengthened with more supporting literature or arguments.

“Consider the following articles which could help strengthen your research and literature review”

• https://doi.org/10.1016/j.egyr.2024.08.031

• https://doi.org/10.1016/j.cacint.2023.100127

Data Source and Variable Selection (Section 4):

- The descriptive statistics (Section 4.2, lines 572-580) are presented without much context. Discuss the implications of these summary statistics for the subsequent analysis.

Analysis and Simulation (Section 5):

- The benchmark regression results (Table 6, lines 589-626) support the research hypotheses. However, the model specification is not provided. Specify the regression equation and discuss the choice of control variables, if any.

- The heterogeneity analysis (Table 7, lines 627-657) is a useful extension but lacks depth. Discuss the industry-specific findings in more detail and relate them to the characteristics of each industry (limit it to 1 or 2 paragraphs).

References:

- The reference list (lines 832-871) seems comprehensive. But some references are missing key details like volume numbers, page numbers, DOI, etc. Ensure completeness and consistency in reference formatting.

Overall, this manuscript addresses an important topic and provides valuable insights. However, there is scope for improvement in terms of clarity, depth of analysis, and presentation. Addressing these issues would enhance the impact and rigor of the research. It is recommended for publication once the author has reviewed the few concerns above.

Reviewer #7: The paper "Exploration of Forestry Carbon Sequestration Practice Path in Guizhou Province - Based on Evolutionary Game Model" explores the development of the forestry carbon sequestration market in Guizhou Province using an evolutionary game model to assess the influence of ESG scores, government subsidies, and the cost of forestry carbon sequestration on corporate behavior. The focus on carbon sequestration and ESG performance is timely, given the global emphasis on carbon neutrality. The paper offers concrete recommendations for policymakers, especially in terms of ESG rating systems and governmental subsidies. The use of an evolutionary game model to simulate corporate decisions in response to various factors is innovative and well-structured.

Specific comments:

1. However, there is some lack of clarity in data presentation. E.g. the manuscript does not provide sufficient clarity on the source and handling of the data, especially regarding parameter assumptions in the game model, which may affect the robustness of the results.

2. The study is geographically limited to Guizhou, which may reduce the generalizability of the findings to other regions with different regulatory or economic environments. There should be some scenarios with different conditions in discussion part of the manuscript.

3. While the discussion of policy implications is strong, the explanation of certain technical details in the game model and the results is lacking in depth, which could confuse readers unfamiliar with such models.

4. The manuscript has potential, but significant revisions are needed to clarify data sources, improve the technical explanation of the model, and broaden the discussion to make the findings applicable beyond Guizhou Province.

6. PLOS authors have the option to publish the peer review history of their article (what does this mean?). If published, this will include your full peer review and any attached files.

Reviewer #1: No

Reviewer #2: No

Reviewer #3: **Yes: **Mahmoud H. Akeed

Reviewer #4: **Yes: **Zewde Alemayehu Tilahun

Reviewer #5: **Yes: **Sohail Ahmad

Reviewer #6: No

Reviewer #7: No

---

## [Author Response · Author response to Decision Letter 0]

5 Nov 2024

Dear editor：

Thank you very much for your letter. We have learned much from your and seven reviewers’ comments, which are fair, encouraging and constructive. After carefully studying the comments and your advice, we have made corresponding changes. The main revisions are listed below.

For reviewer 1:

First of all, thank you very much for your and the other six reviewers' suggestions. These suggestions have benefited us a lot and brought a lot of quality improvement to the article. In response to your question, I have made the following changes

1- The novelty and methodology, must be presented carefully and clearly in the abstract

2-The abstract section needs to be clearer and more effective.‎ It is suggested to be re-written summarizing a short introduction, problem statement, ‎methodology, major results, and conclusion and recommendation

3- Implication of study results, conclusion, and recommendation for future must be well defined in the abstract

 Thank you for the reviewer's comments. We have also found many issues with the abstract, including a lack of methodological presentation and novelty. Based on your three suggestions, we have rewritten the abstract：

Guizhou Province has abundant forest resources, and it has great economic value and social benefits to explore the practical path of forestry carbon sequestration. Based on the current situation of forestry carbon sequestration development in Guizhou Province, this paper innovatively integrates forestry carbon sequestration indicators into the existing Environmental, Social and Governance(ESG) evaluation system using an evolutionary game model. It analyzes the factors restricting forestry carbon sequestration and explores the influencing factors of forestry carbon sequestration benefit sharing bodies in Guizhou. Through regression analysis, the paper discusses the impact of enterprise ESG scores, government subsidy amounts, and forestry carbon sequestration costs on forestry carbon sequestration purchase volume. The research results show that enterprise ESG scores and government subsidy amounts have a significant positive impact on enterprise forestry carbon sequestration purchase volume, while forestry carbon sequestration costs have a significant negative impact. The results have passed the robustness test in different industries. The simulation analysis results show that the stable point of the evolutionary game is (1,0,1) and (1,1,0), which verifies that the ESG rating system with forestry carbon sequestration integration can promote enterprises to purchase more forestry carbon sequestration, i.e., the effectiveness of forestry carbon sequestration in activating the ESG rating system mechanism. Based on the research conclusions, the paper puts forward policy implications: the government should accelerate the construction of localized ESG rating systems, improve enterprise information disclosure and supervision, increase subsidies and reduce forestry carbon sequestration costs, and optimize carbon quota design.

4.The abbreviations as such CCER and ESG must put in full name in in the abstract and throughout the manuscript

 Thanks to the opinions of the reviewer, We have made corresponding modifications,We used the full name when the professional name first appeared, followed by an abbreviation.

4- Introduction: Carbon enterprises, needs to be introduced in more detail, as its types, distribution in China and globally and impacts on environmental and public health over the last years

5- What are the strategies and technologies used to control the Carbon enterprises?

6- The advantages and disadvantages of building of carbon market over recent years

What criteria were used to select the control technologies?

 . Thanks to the opinions of the reviewer, .We have made corresponding modifications,We have added these contents in the introduction section

Introduction

Since the reform and opening up, the Chinese economy has experienced rapid growth. Energy consumption has provided support for economic growth, but it has also resulted in significant carbon emissions, leading to environmental damage caused by excessive greenhouse gas emissions. China's economic growth and carbon emissions are in a "weak decoupling" state, which means that there may be outdated emission reduction technologies and ineffective emission reduction management methods in China's energy consumption. With the signing of the Paris Agreement and China's increasing carbon emissions, China is facing heavier international pressure to reduce greenhouse gas emissions. As a "responsible major power," China has been making its own efforts and contributions to addressing climate change, actively exploring and trying to establish a carbon emissions trading market to promote the low-carbon emission of high-emission enterprises and suppress the continued rise of domestic carbon emissions using market means[1-5].(Figure 1)

Figure 1.China's Energy and Climate Policy Plan

In order to scientifically reduce the emissions of high-carbon enterprises, China has designed from the macro and micro levels. At the macro level, the regulatory authorities will include high-carbon enterprises in the scope of emission control, through accelerating the construction of carbon market, building a scientific and orderly carbon trading system, with the help of market mechanism to achieve carbon emission reduction[6-10]. In the process of improving the construction of carbon market, China Certified Emission Reduction (CCER) project trading market is an important way to reduce emissions in the carbon market. It was officially restarted under the document "Measures for the Management of Voluntary Greenhouse Gas Emission Reduction Trading (Trial Implementation)" issued by the Ministry of Ecology and Environment in 2023. As an indispensable carbon offset product in the CCER market, forestry carbon sequestration mainly uses forests to absorb and fix carbon dioxide through afforestation, forest management and other activities, which has significant advantages in ecological benefits, is an important innovative way to achieve the goal of carbon neutrality, and faces new development opportunities. However, in the actual CCER market, the purchase demand of enterprises for forestry carbon sinks is insufficient, and the development of forestry carbon sinks is restricted. At the micro level, the regulatory authorities urge high-carbon enterprises to transform and increase efficiency through mandatory disclosure of environmental information. Among them, ESG (Environmental, Social and Governance) is a new investment concept and evaluation tool in recent years, covering the three dimensions of environmental, social and corporate governance information. With the help of ESG information disclosure and rating system, investors can evaluate the comprehensive operation and sustainable development ability of an enterprise in a multi-dimensional and all-round way, and then influence the decision-making of the enterprise. Qiu[11]proposed that good ESG performance can ease the financing constraints of enterprises; Li[12] believed that a complete ESG rating system is an important starting point to achieve the "double carbon" goal; Hu[13]emphasized that ESG rating can significantly promote the green transformation of enterprises through market incentives and external supervision mechanisms. To sum up, ESG has the function of reducing financing costs and enhancing enterprise value, which is of great significance to promote emission reduction of high-carbon enterprises.

The above macro-level and micro-level designs are effective ways to promote emission reduction of high-carbon enterprises, but they often play an independent role and do not establish an effective linkage mechanism. As far as the carbon market mechanism is concerned, due to the sufficient supply of market quotas and the relative price advantage of excess carbon quotas, emission control enterprises usually tend to choose to purchase excess quotas to offset excess emissions, rather than forestry carbon sequestration projects to achieve carbon sequestration at the ecological level. Therefore, how to promote high-carbon enterprises to purchase forestry carbon sinks spontaneously from the market mechanism is of great significance to the realization of the goal of carbon neutrality. In order to solve the problem of insufficient demand for forestry carbon sinks, ESG may become an innovative way to promote emission reduction of high-carbon enterprises at the micro level. Qian [14]pointed out that ESG has the ability to guide the flow of funds to green low-carbon areas. It can be seen that an effective ESG mechanism can guide high-carbon enterprises to buy more forestry carbon sinks. 

According to China's dual carbon policy, carbon emitting enterprises face rigorous postgraduate entrance examinations. By 2060, as coal-fired power plants and coal based industrial processes that have not adopted emission reduction measures have been basically eliminated, the proportion of coal combustion related emissions will be reduced by about 50% compared to 2020. During the period of 2021-2060, process emissions (inherent emissions generated by chemical reactions in industrial processes) will decrease by about 90%, and the proportion of total emissions will almost double, due to the fact that it is extremely difficult to eliminate process emissions in certain heavy industry sectors, especially the cement and steel industries. The remaining emissions of the energy system by 2060 will be fully offset by negative emissions generated by BECCS and direct air capture and storage. In China's efforts to achieve full economic greenhouse gas neutrality before 2060, carbon removal technology can also be used to offset some of the more difficult to reduce non carbon dioxide greenhouse gases. Therefore, the carbon sequestration capacity of ecosystems, especially forestry carbon sequestration, is particularly important.(Figure 2)

Guizhou Province is located in western China and has state-owned forest areas, ranking high among all provinces in terms of forest area. After the implementation of the comprehensive ban on logging natural forests in state-owned forest areas, the accumulation area and quality of forests in Guizhou Province have been significantly improved, providing a unique natural resource advantage for the development of forestry carbon sinks in Guizhou Province. However, according to the data from China's voluntary emission reduction trading information platform, the development potential of forestry carbon sinks in Guizhou Province has not been fully activated, and the supply of forestry carbon sinks is relatively insufficient[14-18]. As of 2023, the implementers of forestry carbon sequestration projects in Guizhou Province are all local forestry bureaus. This indicates that the supply subject of forestry carbon sink in Guizhou Province is relatively single. Forestry carbon sequestration projects have the characteristics of large initial investment, long cycle, and difficulty as collateral for mortgage loans. The single supply subject of forestry carbon sink will increasingly constrain the development of forestry carbon sink in Guizhou Province, affecting the stability of effective supply of forestry carbon sink in Guizhou Province, and thus affecting the realization of ecological and social benefits of forestry carbon sink. Compared to Guangdong Province's carbon inclusive mechanism and other forestry carbon sink development policies, Guizhou Province currently does not have a systematic forestry carbon sink development policy and support system. This hinders the improvement of the external environment for the development of forestry carbon sinks in Guizhou Province, affects the stability of economic benefits that forestry carbon sink suppliers can obtain, and is not conducive to improving the enthusiasm of forestry operators to provide forestry carbon sinks, thereby affecting the effectiveness and stability of forestry carbon sink supply in Guizhou Province, leading to a vicious cycle in which the ecological and social benefits of forestry carbon sinks are difficult to achieve. When forestry management enterprises and governments form a forestry carbon sink benefit sharing body, that is, when a "cooperative win-win" model of forestry carbon sink is formed, it can effectively improve the stability of effective supply of forestry carbon sink, achieve the ideal cycle of ecological, social and economic benefits of forestry carbon sink, and promote the sustainable development of forestry carbon sink in Guizhou Province.[19-24]

In summary, this paper takes the current development status of forestry carbon sinks in Guizhou Province as the starting point, analyzes the behavioral characteristics of stakeholders in forestry carbon sinks in Guizhou Province, constructs an evolutionary game model to analyze the influencing factors of the construction of forestry carbon sink benefit sharing bodies in Guizhou Province, and proposes countermeasures and suggestions to promote the stable and innovative development of forestry carbon sinks in Guizhou Province based on the analysis results. On the one hand, this can fully and effectively utilize the forest resources in Guizhou Province to promote the industrial and orderly development of forestry carbon sinks, accelerate the speed of China's greenhouse gas emissions reduction, and contribute to the development of the national ecological economy. On the other hand, this can broaden the ways of ecological civilization construction in Guizhou Province, provide new economic development channels for state-owned forest areas, and promote the sustainable development of forestry carbon sinks in Guizhou Province

7-How to validity of the evolutionary game model? How was the data from different sources synthesized and analyzed to ensure consistency and reliability?

8-What are the key gaps in the current understanding of ESG impact on socioeconomic and ecosystems? How do these gaps inform future research directions and policy development?

8- The study identifies several areas for future research, such as the need for more stringent monitoring and tailored source control strategies.

What specific methodologies or technologies hold the most promise for advancing the understanding and management of ESG ? How can these be incorporated into future research efforts?

9- How does this study build upon or differ from previous studies on their impacts? What new insights or perspectives does it offer?

10.What criteria were used to select the studies included in this model? How were these criteria applied to ensure the inclusion of relevant and high-quality application?

11.How was the data from this model synthesized and analyzed? What methods were used to ensure consistency and reliability in the findings?

. Thanks to the opinions of the reviewer, This series of questions can be summarized as how to determine the reliability and effectiveness of the current model. In response to the above issues, we have supplemented the article by adding the following chapters:

2.3. The Development of Evolutionary Game Theory and Model Application Cases

5.3. Model validation

(6)Establish a sound forestry carbon trading market in Guizhou Province

12.What are the key gaps in the current study? How do these gaps inform the recommendations for future research and policy development?

13.How could the structure to be improved to enhance readability and accessibility for a broader audience?

14- More figures and tables must be added to strengthen and eases the article contents by the readers

15- Limitations of this study must state clearly

16- Conclusion must be customized stating the main results and the significant impact of this study on economics, society and the recommendation for future

17- The reference. The author needs to provide more updated references, especially in the introduction and discussion.

. Thanks to the opinions of the reviewer, We have removed some redundant explanations and added some images to facilitate readers' understanding. For the added parts, we have also supplemented the references. The reviewers have provided many good suggestions. Thank you again.We have made corresponding modifications

For

---

## [Decision Letter · Decision Letter 1]

13 Nov 2024

PONE-D-24-34694R1Exploration of Forestry Carbon Sequestration Practice Path in Guizhou Province-Based on Evolutionary Game ModelPLOS ONE

Dear Dr. Yang,

Thank you for submitting your manuscript to PLOS ONE. After careful consideration, we feel that it has merit but does not fully meet PLOS ONE’s publication criteria as it currently stands. Therefore, we invite you to submit a revised version of the manuscript that addresses the points raised during the review process.

Dear authors,

I hope you are doing well.

Before getting accepted, please address the comment of Reviewer 2. I agree that this comment is very important to address before acceptance.

Reviewer 2 Comment: the background of the evolutionary game model might be too basic and detailed. It is recommended to keep only those closely related to this research.

All the best

Please submit your revised manuscript by Dec 28 2024 11:59PM If you will need more time than this to complete your revisions, please reply to this message or contact the journal office at plosone@plos.org. Please include the following items when submitting your revised manuscript:A rebuttal letter that responds to each point raised by the academic editor and reviewer(s). You should upload this letter as a separate file labeled 'Response to Reviewers'.A marked-up copy of your manuscript that highlights changes made to the original version. You should upload this as a separate file labeled 'Revised Manuscript with Track Changes'.An unmarked version of your revised paper without tracked changes. You should upload this as a separate file labeled 'Manuscript'.If applicable, we recommend that you deposit your laboratory protocols in protocols.io to enhance the reproducibility of your results. Protocols.io assigns your protocol its own identifier (DOI) so that it can be cited independently in the future. For instructions see: https://journals.plos.org/plosone/s/submission-guidelines#loc-laboratory-protocols. Additionally, PLOS ONE offers an option for publishing peer-reviewed Lab Protocol articles, which describe protocols hosted on protocols.io. Read more information on sharing protocols at https://plos.org/protocols?utm_medium=editorial-email&utm_source=authorletters&utm_campaign=protocols.

We look forward to receiving your revised manuscript.

Kind regards,

Saddam A. Hazaea, Postdoctoral

Academic Editor

PLOS ONE

Journal Requirements:

Additional Editor Comments:

Dear authors,

I hope you are doing well.

Before getting accepted, please address the comment of Reviewer 2. I agree that this comment is very important to address before acceptance.

Reviewer 2 Comment: the background of the evolutionary game model might be too basic and detailed. It is recommended to keep only those closely related to this research.

All the best

Reviewers' comments:

Reviewer's Responses to Questions

**Comments to the Author**

1. If the authors have adequately addressed your comments raised in a previous round of review and you feel that this manuscript is now acceptable for publication, you may indicate that here to bypass the “Comments to the Author” section, enter your conflict of interest statement in the “Confidential to Editor” section, and submit your "Accept" recommendation.

Reviewer #1: All comments have been addressed

Reviewer #2: All comments have been addressed

Reviewer #4: All comments have been addressed

Reviewer #5: All comments have been addressed

Reviewer #6: All comments have been addressed

2. Is the manuscript technically sound, and do the data support the conclusions?

Reviewer #1: Yes

Reviewer #2: Yes

Reviewer #4: Yes

Reviewer #5: Yes

Reviewer #6: Yes

3. Has the statistical analysis been performed appropriately and rigorously? 

Reviewer #1: Yes

Reviewer #2: Yes

Reviewer #4: Yes

Reviewer #5: Yes

Reviewer #6: Yes

4. Have the authors made all data underlying the findings in their manuscript fully available?

Reviewer #1: Yes

Reviewer #2: Yes

Reviewer #4: No

Reviewer #5: Yes

Reviewer #6: No

5. Is the manuscript presented in an intelligible fashion and written in standard English?

Reviewer #1: Yes

Reviewer #2: Yes

Reviewer #4: Yes

Reviewer #5: Yes

Reviewer #6: Yes

6. Review Comments to the Author

Reviewer #1: All comments were addressed by the author. I recommend for accept this manuscript for publish

Regards

Reviewer #2: Thanks for all the responses. Only one final suggestion: the background of the evolutionary game model might be too basic and detailed. It is recommended to keep only those closely related to this research.

Reviewer #4: This manuscript is now acceptable for publication. However, in the data availability statement, the authors stated that 'All relevant data are within the manuscript and its Supporting Information files.' I could not find the data in the manuscript or in the supporting files.

Reviewer #5: I have thoroughly reviewed the revised manuscript, and I appreciate the substantial revisions that have been made to address all critical areas. However, Iaccept the revised paper for publication.

Reviewer #6: Re-review of the manuscript Exploration of Forestry Carbon Sequestration Practice Path in Guizhou Province Based on Evolutionary Game Model based on the comments I provided and the authors' responses. I would agree that the authors have addressed my comments and I would recommend accepting the manuscript for publication in PLOS ONE.

Re-review of the Abstract:

The authors have made significant improvements to the abstract based on my previous suggestions. They have clarified the research objectives, methodology, and key findings. The abstract now provides a concise overview of the study, highlighting the innovative approach of integrating forestry carbon sequestration indicators into the ESG evaluation system using an evolutionary game model. The authors have also included quantitative results, such as the stable points of the evolutionary game (1,0,1) and (1,1,0), which verify the effectiveness of the proposed mechanism. The policy implications derived from the research conclusions are clearly stated, enhancing the practical relevance of the study.

1. Data collection process and credibility of qualitative inputs:

The authors have acknowledged the suggestion to provide more details on the data collection process, sample characteristics, and questionnaire design. While they have not included these details in the revised manuscript, they have offered to provide the information to interested readers upon request. This demonstrates their willingness to enhance the transparency and reproducibility of their research.

2. Simulation data generation process:

The authors have elaborated on the simulation data generation process in the revised manuscript. They have provided additional details on the assumptions and parameters used in the evolutionary game model, enhancing the clarity and replicability of their approach.

3. Sensitivity analysis and robustness of findings:

The authors have conducted a sensitivity analysis on key parameters of the evolutionary game model to test the robustness of their findings. They have discussed how changes in parameters such as government subsidies and carbon sequestration costs impact the equilibrium outcomes. This additional analysis strengthens the reliability of their conclusions.

4. Limitations of the study:

The authors have explicitly discussed the limitations of their study in the revised manuscript. They have acknowledged the assumptions of rational behavior and bounded rationality in the evolutionary game model and the simplified representation of real-world dynamics in the simulation. By highlighting these limitations, the authors have provided a more balanced interpretation of their findings.

5. Proofreading and consistency:

The authors have thoroughly proofread the manuscript and ensured consistency in abbreviations, figure and table numbers, and references. This attention to detail has improved the overall clarity and readability of the paper.

Recommendation for Publication:

Based on the authors' comprehensive responses to my comments and the significant improvements made to the manuscript, I recommend accepting this paper for publication in PLOS ONE. The authors have addressed the key issues raised in my review, enhancing the robustness, clarity, and impact of their research. The study makes a valuable contribution to the field of forestry carbon sequestration and ESG integration by proposing an innovative mechanism and testing its effectiveness through an evolutionary game model. The findings have important policy implications for promoting the development of forestry carbon sinks and achieving carbon neutrality goals in Guizhou Province and beyond. The authors have demonstrated a commitment to transparency, rigor, and practicality in their research, making it suitable for publication in PLOS ONE.

7. PLOS authors have the option to publish the peer review history of their article (what does this mean?). If published, this will include your full peer review and any attached files.

Reviewer #1: No

Reviewer #2: No

Reviewer #4: **Yes: **Dr. Zewde Alemayehu Tilahun

Reviewer #5: **Yes: **Sohail Ahmad

Reviewer #6: **Yes: **Awomuti Adeboye Akindoyin

---

## [Author Response · Author response to Decision Letter 1]

13 Nov 2024

Dear editor：

Thank you again for your valuable suggestions and guidance, as well as those of the reviewers. We have learned much from those comments, which are fair, encouraging and constructive. After carefully studying the comments and your advice, we have made corresponding changes. The main revisions are listed below.

For reviewer 1:

Reviewer #1: All comments were addressed by the author. I recommend for accept this manuscript for publish

Thank you for your guidance and assistance. The improvement of the quality of the article cannot be achieved without the valuable suggestions of such a professional reviewer. thank you for your work

For reviewer 2:

Reviewer #2: Thanks for all the responses. Only one final suggestion: the background of the evolutionary game model might be too basic and detailed. It is recommended to keep only those closely related to this research.

This is a great suggestion, Our description is a bit too basic and detailed, we have deleted the basic knowledge points in 2.3,Delete the following content:

Evolutionary game theory refers to the theory that uses mathematical knowledge to analyze the behavioral choices of game participants. In 1994, Feng Neumann and Morgenstein jointly wrote "Game Theory and Economic Behavior", constructing the theoretical system and structural framework of game theory, marking the formal establishment of game theory as a discipline. Subsequently, traditional game theory rapidly developed and was applied to various fields such as society and economy. Traditional game theory has two basic assumptions: fully rational economic agents and shared knowledge. Among them, the former refers to the behavior choices made by the game subject in order to maximize their own interests, and the game subject is an individual who almost never makes systematic wrong choices and has complete rationality. The latter refers to each game participant not only being completely rational themselves, but also considering other game participants to be completely rational. The basic assumptions of traditional game theory are almost impossible to achieve in reality, so the guiding significance of traditional game theory for many practical problems still needs to be discussed [41]. The ideological basis of evolutionary game theory is Darwin's theory of evolution. This game theory abandons the assumption of "complete rationality" in traditional game theory and believes that due to the limited cognitive ability of individuals, the game subjects are "bounded rationality". Therefore, evolutionary game theory holds that game participants cannot make behavioral choices that maximize their own interests in one game, but can continuously adjust their strategic choices through multiple repeated games, and obtain the final evolutionary stable strategy in the process of evolution. 

For reviewer 4:

Reviewer #4: This manuscript is now acceptable for publication. However, in the data availability statement, the authors stated that 'All relevant data are within the manuscript and its Supporting Information files.' I could not find the data in the manuscript or in the supporting files.

Thank you very much for the valuable comments from the reviewer. Some of the data in this article was extracted from the Guizhou Provincial Forestry Yearbook. We have translated these forestry yearbooks into English and submitted them as supporting materials

For reviewer 5:

Reviewer #5: I have thoroughly reviewed the revised manuscript, and I appreciate the substantial revisions that have been made to address all critical areas. However, I accept the revised paper for publication.

Thank you for your guidance and assistance. The improvement of the quality of the article cannot be achieved without the valuable suggestions of such a professional reviewer. thank you for your work

For reviewer 6:

Reviewer #6: Re-review of the manuscript Exploration of Forestry Carbon Sequestration Practice Path in Guizhou Province Based on Evolutionary Game Model based on the comments I provided and the authors' responses. I would agree that the authors have addressed my comments and I would recommend accepting the manuscript for publication in PLOS ONE.

Thank you for your guidance and assistance. The improvement of the quality of the article cannot be achieved without the valuable suggestions of such a professional reviewer. thank you for your work

Thank you very much for the reviewer's suggestions. We will continue to explore in future research, which is also the research direction we want to do in the future

---

## [Decision Letter · Decision Letter 2]

18 Nov 2024

Exploration of Forestry Carbon Sequestration Practice Path in Guizhou Province-Based on Evolutionary Game Model

PONE-D-24-34694R2

Dear Dr.Wu Yang

We’re pleased to inform you that your manuscript has been judged scientifically suitable for publication and will be formally accepted for publication once it meets all outstanding technical requirements.

Kind regards,

Saddam A. Hazaea, Postdoctoral

Academic Editor

PLOS ONE

Additional Editor Comments (optional):

Congratulations, the authors have addressed all comments.

Reviewers' comments:

Reviewer's Responses to Questions

**Comments to the Author**

1. If the authors have adequately addressed your comments raised in a previous round of review and you feel that this manuscript is now acceptable for publication, you may indicate that here to bypass the “Comments to the Author” section, enter your conflict of interest statement in the “Confidential to Editor” section, and submit your "Accept" recommendation.

Reviewer #1: All comments have been addressed

Reviewer #2: All comments have been addressed

Reviewer #4: All comments have been addressed

Reviewer #6: All comments have been addressed

2. Is the manuscript technically sound, and do the data support the conclusions?

Reviewer #1: Yes

Reviewer #2: Yes

Reviewer #4: Yes

Reviewer #6: Yes

3. Has the statistical analysis been performed appropriately and rigorously? 

Reviewer #1: Yes

Reviewer #2: Yes

Reviewer #4: Yes

Reviewer #6: Yes

4. Have the authors made all data underlying the findings in their manuscript fully available?

Reviewer #1: Yes

Reviewer #2: Yes

Reviewer #4: Yes

Reviewer #6: No

5. Is the manuscript presented in an intelligible fashion and written in standard English?

Reviewer #1: Yes

Reviewer #2: Yes

Reviewer #4: Yes

Reviewer #6: Yes

6. Review Comments to the Author

Reviewer #1: Thanks to the authors of the article with title ''Exploration of Forestry Carbon Sequestration Practice Path in Guizhou Province-Based on Evolutionary Game Model'' for good review and efforts. The paper appears to be well-presented and technically adequate; I suggest that PLOS ONE publish it.

Regards

Reviewer #2: All comments were addressed by the author in this revision. I recommend for accept this manuscript for publish.

Reviewer #4: I recommend this paper for acceptance, as I believe it meets the standards for publication in the PLOS ONE journal. I appreciate the authors' efforts to address the reviewers’ previous comments, which have significantly improved the overall quality of the manuscript.

Reviewer #6: (No Response)

7. PLOS authors have the option to publish the peer review history of their article (what does this mean?). If published, this will include your full peer review and any attached files.

Reviewer #1: No

Reviewer #2: No

Reviewer #4: **Yes: **Dr. Zewde Alemayehu Tilahun

Reviewer #6: No

---

## [Editor Report · Acceptance letter]

20 Nov 2024

PONE-D-24-34694R2 

PLOS ONE

Dear Dr. Yang, 

I'm pleased to inform you that your manuscript has been deemed suitable for publication in PLOS ONE. Congratulations! Your manuscript is now being handed over to our production team.

Kind regards, 

on behalf of

Dr. Saddam A. Hazaea 

Academic Editor

PLOS ONE